# Separable actions of acetylcholine and noradrenaline on neuronal ensemble formation in hippocampal CA3 circuits

**Luke Y. Prince**[1,2,3], **Travis Bacon**[1], **Rachel Humphries**[1], **Krasimira Tsaneva-Atanasova**[4,5], **Claudia Clopath**[6], **Jack R. Mellor**[1] *

**1** Centre for Synaptic Plasticity, School of Physiology Pharmacology, and Neuroscience, University of Bristol, Bristol, United Kingdom, **2** Mila, Montreal, Quebec, Canada, **3** School of Computer Science, McGill University, Montreal, Quebec, Canada, **4** Department of Mathematics and Living Systems Institute, University of Exeter, Exeter, United Kingdom, **5** EPRSC Centre for Predictive Modelling in Healthcare, University of Exeter, Exeter, United Kingdom, **6** Bioengineering Department, Imperial College London, London, United Kingdom

* Jack.Mellor@Bristol.ac.uk

**Data Availability Statement:** Computational modelling code for all simulations is available at https://github.com/lyprince/mossy-fibre-ca3-ach-

## Abstract

In the hippocampus, episodic memories are thought to be encoded by the formation of ensembles of synaptically coupled CA3 pyramidal cells driven by sparse but powerful mossy fiber inputs from dentate gyrus granule cells. The neuromodulators acetylcholine and noradrenaline are separately proposed as saliency signals that dictate memory encoding but it is not known if they represent distinct signals with separate mechanisms. Here, we show experimentally that acetylcholine, and to a lesser extent noradrenaline, suppress feed-forward inhibition and enhance Excitatory–Inhibitory ratio in the mossy fiber pathway but CA3 recurrent network properties are only altered by acetylcholine. We explore the implications of these findings on CA3 ensemble formation using a hierarchy of models. In reconstructions of CA3 pyramidal cells, mossy fiber pathway disinhibition facilitates postsynaptic dendritic depolarization known to be required for synaptic plasticity at CA3-CA3 recurrent synapses. We further show in a spiking neural network model of CA3 how acetylcholine-specific network alterations can drive rapid overlapping ensemble formation. Thus, through these distinct sets of mechanisms, acetylcholine and noradrenaline facilitate the formation of neuronal ensembles in CA3 that encode salient episodic memories in the hippocampus but acetylcholine selectively enhances the density of memory storage.

## Author summary

How the brain decides which experiences to encode to memory and which to discard is a fundamental question in neuroscience. The neuromodulators acetylcholine and noradrenaline are believed to separately play a central role in determining what is encoded but the mechanisms by which they act are mostly unknown and there have been no direct comparisons made between these two critical neuromodulators. In this study, we investigate the effects of acetylcholine and noradrenaline on a key circuit responsible for the

na/tree/draft. All other relevant data are within the manuscript and its Supporting Information files.

**Funding:** LYP, RH, CC and JRM were funded by the Wellcome Trust (101029/Z/13/Z grant to JRM). TB and JRM were funded by the Biotechnology and Biological Sciences Research Council (BBSRC, BB/R002177/1 grant to JRM). CC was funded by an IBRO & Simons Grant (ID # isiCNI2017). KTA was funded by the Engineering and Physical Sciences Research Council (EPSRC, EP/T017856/1). The funders had no role in study design, data collection and analysis, decision to publish, or preparation of the manuscript.

encoding of memories, namely, the dentate gyrus–CA3 microcircuit in the hippocampus. Using slice electrophysiology, we measure the effects of acetylcholine and noradrenaline on key synaptic and cellular nodes within this neuronal network. We then explore the network level implications of these findings on neuronal ensemble formation using a hierarchy of computational models. Based on the observed physiological effects of acetylcholine and noradrenaline, our models predict that acetylcholine facilitates efficient formation of ensembles within CA3 with a high degree of overlap whereas noradrenaline has more limited effects and no impact on the efficiency or overlap of ensemble formation.

## Introduction

The hippocampus plays a central role in the formation of episodic memories by processing information from the entorhinal cortex sequentially through the dentate gyrus, CA3 and CA1 regions. Anatomical, functional and theoretical considerations propose separate computational properties for each of these regions in support of memory processing. In particular, the CA3 region is characterized by a recurrently connected set of excitatory pyramidal neurons, which are believed to encode auto-associative memories by selectively strengthening recurrent synapses between ensembles of neurons that provide a neural representation of the memory [1,2]. Configuring a recurrent network in this way is proposed to endow the network with attractor dynamics, in which the network is driven towards a stable state of ensemble formation [3–6]. This process is related to memory retrieval, in which external sources of input will alter the state of the network by activating subsets of neurons and through recurrent dynamics will be driven towards these stable states in which all neurons in the ensemble are reactivated– a process also referred to as pattern completion [7,8]. Within this framework, memory encoding is believed to be the procedure of altering the network through synaptic plasticity to create or change the position of attractor states [9,10]. However, not all memories are stored, indicating that there may be a gate to select which experiences should be encoded, but it is unclear how such a filter might operate.

One potential filter mechanism is the release of neuromodulators such as acetylcholine or noradrenaline that can rapidly reconfigure neuronal networks [11–14]. Since acetylcholine and noradrenaline are both released in response to salient, rewarding or arousing stimuli that require learning of new associations [12,13,15–19] it is proposed that each should facilitate the formation of new memory ensembles in the hippocampus. Commensurate with this, acetylcholine facilitates NMDA receptor function and induction of synaptic plasticity [20–26] and selectively suppresses recurrent activity representing previously stored information in favour of feed-forward activity representing novel information [27,28]. These properties are predicted to facilitate the encoding of new memories and allow greater overlap between representations [11,28]. Similarly, noradrenaline enhances cellular excitability and facilitates the induction of synaptic plasticity to shift hippocampal representations [18,29–31]. However, differences between the actions of acetylcholine and noradrenaline are also proposed where acetylcholine release corresponds to states of expected uncertainty requiring limited update in existing memory representations and noradrenaline release corresponds to unexpected uncertainty requiring a state shift [32,33]. Discerning between these different conceptual outcomes requires direct comparison of the mechanisms engaged by each neuromodulator.

The dentate gyrus receives excitatory glutamatergic input from layer II of the medial entorhinal cortex, and sparsifies this signal by suppressing the activity of most dentate gyrus granule cells through lateral inhibition while dramatically increasing the firing rate of a select few

granule cells [34,35]. By this mechanism granule cells detect salient, novel information and accentuate minor contextual details related to familiar information (a process often referred to as pattern separation). Individual granule cells provide a strong, sparse, facilitating input to a small number of CA3 pyramidal cells that can be sufficiently powerful to engage 1:1 spike transfer after multiple spikes in a granule cell burst [36–40]. This focal excitation by mossy fibers drives synchronisation between subsets of CA3 pyramidal cells allowing recurrent CA3-CA3 synapses to engage Hebbian plasticity mechanisms to create ensembles of strongly coupled CA3 cells thereby initiating the storage of new information [10,35,41–44]. Mossy fibers also excite a broad and diverse set of inhibitory interneurons that provide a widespread 'blanket' of feed-forward inhibition over a large population of CA3 pyramidal cells [40,45–47]. This feed-forward inhibition prevents runaway excitation and ensures tight spike timing for spike transfer [36,48,49], while also enhancing memory precision [50] but it is not known what impact it may have on the conditions required for synaptic plasticity within the CA3 recurrent network and how neuromodulation regulates Excitatory-Inhibitory balance.

Here we compare the modulation by acetylcholine or noradrenaline of ensemble creation and therefore memory encoding within the hippocampal CA3 network. We investigate these neuromodulator effects on 3 critical features of ensemble creation: i) the ability for mossy fiber input to create the conditions necessary for synaptic plasticity at recurrent CA3 pyramidal neuron connections, ii) the excitability of CA3 pyramidal neurons, and iii) the overall strength of CA3 recurrent connectivity [51]. Using slice electrophysiology and a hierarchy of experimentally constrained computational models of mossy fiber synaptic transmission and CA3 network activity, we demonstrate that acetylcholine, and to a lesser extent noradrenaline, suppress feed-forward inhibition creating the conditions to enable plasticity at recurrent CA3-CA3 synapses and the formation of ensembles within the CA3 network. Furthermore, we show that acetylcholine, but not noradrenaline, increases the excitability of CA3 pyramidal neurons and reduces their overall connectivity thereby increasing the density of stable ensembles by enhancing permissible overlap between ensembles.

## Results

A critical feature of ensemble formation in CA3 is the mossy fiber input from dentate gyrus. However, the cumulative effects of acetylcholine or noradrenaline on the mossy fiber projection incorporating both excitatory and inhibitory synaptic transmission are not known. Therefore, we first recorded experimentally the effect of acetylcholine or noradrenaline on combined feed-forward excitatory and inhibitory synaptic transmission in the mossy fiber pathway of mouse hippocampal slices (Experimental setup: Fig 1A). Minimal stimulation of granule cells resulted in EPSCs and IPSCs that were individually isolated by setting the membrane potential to -70mV and +10mV respectively in accordance with experimentally determined reversal potentials for inhibitory and excitatory transmission respectively (S1C and S1D Fig). Mossy fibers were stimulated with trains of 4 pulses at 20 Hz every 20 s, in accordance with previous studies of mossy fiber short-term plasticity [49,52,53]. Application of the group II mGluR agonist DCG-IV (1 μM) reduced EPSC amplitudes by >90% (Fig 1B, left) indicating selective activation of the mossy fiber pathway [54]. The EPSC rise times (20–80%), latencies, and jitter were $0.57 \pm 0.11$ ms, $2.1 \pm 0.49$ ms and $0.53 \pm 0.27$ ms respectively (S1A Fig) characteristic of mossy fiber synapses and confirming their monosynaptic origin [55]. DCG-IV also reduced IPSC amplitudes by >90% (Fig 1B, right) and the rise times, latencies and jitter were $3.52 \pm 1.08$ ms, $6.20 \pm 1.82$ ms and $0.73 \pm 0.25$ ms respectively (S1B Fig) indicating that IPSCs were mediated by disynaptic feed-forward inhibitory transmission in the mossy fiber pathway

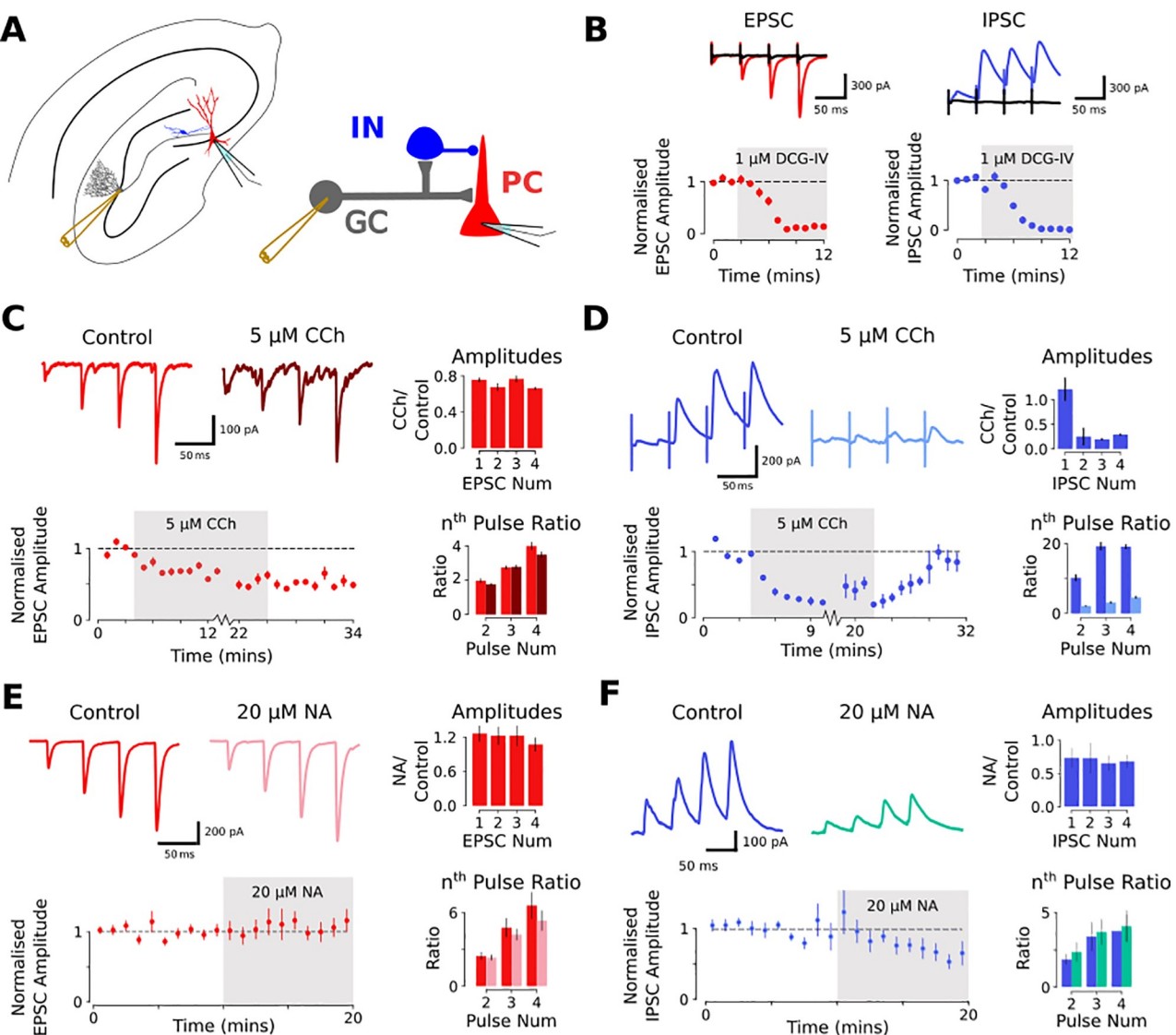

**Fig 1. The effects of carbachol and noradrenaline on feed-forward excitatory and inhibitory transmission in the mossy fiber pathway.** A) Left: Experimental setup indicating the location of stimulation and recording electrodes within a hippocampal slice. Right: Schema of feed-forward mossy fiber circuit. Granule cells (GC) in the dentate gyrus send mossy fiber axons to synapse onto feed-forward interneurons (IN) and CA3 pyramidal cells (PC). B) EPSCs and IPSCs evoked by granule cells stimulation were blocked by 1 μM DCG-IV, confirming responses were driven by mossy fiber activation. Top: Example traces of DCG-IV block (black) of EPSCs (red) and IPSCs (blue). Bottom: Time course of DCG-IV block of 4th EPSC (red, n = 9) and IPSC (blue, n = 6). C) 5 μM CCh mildly suppresses mossy fiber EPSCs. Top left: Example traces before and after bath application of 5 μM CCh. Bottom Left: Time course of CCh effect, and washout (n = 7). Top Right: Effect of CCh on response amplitudes for each pulse. Bottom Right: Effect of CCh on nth/1st Pulse Ratio. D) 5 μM CCh substantially reduces disynaptic mossy fiber driven IPSC amplitudes. Top left: Example traces before and after bath application of 5 μM CCh. Bottom Left: Time course of CCh effect, and washout (n = 5). Top Right: Effect of CCh on response amplitudes for each pulse. Bottom Right: Effect of CCh on nth/1st Pulse Ratio. E) 20 μM NA has no effect on mossy fiber EPSCs. Top left: Example traces before and after bath application of 20 μM NA. Bottom Left: Time course of NA effect (n = 7). Top Right: Effect of NA on response amplitudes for each pulse. Bottom Right: Effect of NA on nth/1st Pulse Ratio. F) 20 μM NA reduces disynaptic mossy fiber driven IPSC amplitudes. Top left: Example traces before and after bath application of 20 μM NA. Bottom Left: Time course of NA effect (n = 5). Top Right: Effect of NA on response amplitudes for each pulse. Bottom Right: Effect of NA on nth/1st Pulse Ratio.

[49]. Both EPSCs and IPSCs exhibited pronounced facilitation in response to a train of 4 stimuli at 20 Hz (Fig 1B and 1F) as previously shown for mossy fiber feed-forward excitatory and inhibitory pathways [49].

## Acetylcholine and noradrenaline reduce feed-forward inhibition and alter short-term plasticity in the mossy fiber pathway

The impact of acetylcholine or noradrenaline on information transfer between the dentate gyrus and CA3 will depend on its effects on both excitatory and inhibitory pathways. To assess the effect of acetylcholine on both pathways we used the broad-spectrum cholinergic receptor agonist carbachol (CCh). Application of 5 μM CCh depressed EPSC amplitudes by ~25% (Fig 1C; 75.6 ± 17.0% and 66.2 ± 11.9% of baseline measured at 1$^{st}$ and 4th pulses, n = 7, $p < 0.05$) without altering facilitation ratios (Fig 1C; $p = 0.509$; Measured at 1$^{st}$ to 4$^{th}$ pulse). This depression did not recover on washout of CCh indicating a form of muscarinic receptor-induced long-term depression [56]. The use of minimal stimulation meant that responses to the first stimuli were highly variable and often very small or absent due to the low basal probability of release at mossy fiber synapses [38,49,55]. This was particularly true for IPSC recordings resulting in very large facilitation ratios and a highly variable effect of CCh on the first IPSC in a train. In contrast to the effect on EPSCs, 5 μM CCh had no consistent effect on the 1$^{st}$ IPSC in a train but depressed subsequent IPSC amplitudes reversibly and to a much greater degree (Fig 1D; 121.5 ± 19.5% and 29.3 ± 13.5% of baseline measured at 1$^{st}$ and 4$^{th}$ pulses; n = 6, $p = 0.012$ measured at 4$^{th}$ pulse) and at the same time reduced facilitation ratios (Fig 1D; $p = 0.012$; measured at 1$^{st}$ to 4$^{th}$ pulse). Because the larger IPSCs were greatly reduced by CCh this represents a substantial reduction in feed-forward inhibition across the 4-pulse stimulus train. CCh also enhances the excitability of neurons in the CA3 network [28,57,58] but this effect was absent at a cellular level in our recordings because of the inclusion of cesium in the pipette solution (S1G Fig). Elevated network excitability was evident from a general increase in the frequency of spontaneous EPSCs and IPSCs (S1E and S1F Fig). At lower concentrations, 1 μM CCh had limited effect on IPSC amplitudes whereas at higher concentrations 10 μM CCh had similar effects to 5 μM CCh with a substantial depression of IPSC amplitudes (S1I Fig). These results indicate that acetylcholine causes a small depression of excitatory transmission at mossy fiber synapses whereas feed-forward inhibitory transmission is substantially depressed. Overall, this indicates a substantial net enhancement of Excitatory-Inhibitory ratio in the mossy fiber pathway in the presence of acetylcholine.

To assess the effect of noradrenaline on both pathways we used bath application of 20 μM noradrenaline (NA) that we have found to be a maximal effective dose for cellular and synaptic properties within hippocampal slices [31]. Application of 20 μM noradrenaline had no effect on EPSC amplitudes (Fig 1E; 126.1 ± 13.9% and 107.1 ± 12.1% of baseline measured at 1$^{st}$ and 4th pulses, n = 14, $p = 0.44$) or facilitation ratios (Fig 1E; $p = 0.08$; Measured at 1$^{st}$ to 4$^{th}$ pulse). In contrast to the effect on EPSCs, 20 μM NA depressed IPSC amplitudes by ~30% (Fig 1F; 73.3 ± 14.7% and 68.1 ± 9.8% of baseline measured at 1$^{st}$ and 4$^{th}$ pulses; n = 10, $p = 0.04$ measured at 4$^{th}$ pulse) without altering facilitation ratios (Fig 1F; $p = 0.74$; measured at 1$^{st}$ to 4$^{th}$ pulse). This represents a smaller inhibition of feed-forward inhibition in comparison to CCh. These results indicate that the separate effects of noradrenaline on feed-forward excitatory or inhibitory synaptic transmission in the mossy fiber pathway are very different to acetylcholine. However, the combined net enhancement of Excitatory-Inhibitory ratio is potentially similar for noradrenaline and acetylcholine.

Information transfer between the dentate gyrus and CA3 network depends on bursts of high frequency activity in dentate granule cells leading to pronounced frequency facilitation of excitatory synaptic input [36,37,39]. This is balanced by frequency-dependent facilitation of inhibitory synaptic input [49] but variations in the short-term plasticity dynamics between the excitatory and inhibitory pathways will lead to windows within the frequency domain when excitation dominates and action potentials are triggered in CA3 pyramidal cells [59]. However,

these temporal windows have not been fully characterized and, furthermore, the effect of acetylcholine or noradrenaline on Excitatory-Inhibitory ratio over a range of mossy fiber stimulation patterns is not known. To investigate the patterns of activity that trigger action potentials under conditions of presence and absence of acetylcholine or noradrenaline we adapted a Tsodyks-Markram based model of short-term plasticity dynamics in both excitatory and inhibitory pathways (see Materials and Methods).

Short-term plasticity models are difficult to constrain with responses evoked by regular stimulation protocols [60]. Therefore, we constrained the model using responses to a stimulation pattern resembling the natural spike statistics of dentate gyrus granule cells which incorporate a broad range of inter-stimulus intervals (ISIs) (Fig 2A) [52,61] whose distribution is similar to multiple reports of granule cell activity (e.g. [62]). Similar to the regular stimulation pattern of 4 stimuli at 20Hz, CCh depressed EPSCs and IPSCs in response to the irregular stimulation pattern across the range of ISIs but the depression was much more pronounced for IPSCs (Fig 2B and 2C). Likewise, noradrenaline had little effect on EPSCs and produced a small depression of IPSCs (Fig 2B and 2C). Several phenomenological short-term plasticity models of increasing level of complexity for both excitatory and inhibitory synaptic responses were assessed for fit to the experimental data (see Materials and Methods for detailed description of these models). The basic form of these models included a facilitation and a depression variable, here represented as $f$ and $d$ respectively. Dynamics for these variables are governed by parameters for degree ($a$) and timecourse of facilitation ($\tau_f$) and depression ($\tau_d$) as well as baseline of release ($f_0$) and synaptic conductance ($g$) (Fig 2D) that correspond to presynaptic neurotransmitter release processes that vary between synapse types. Parameter inference for the short-term plasticity models was carried out using Markov-Chain Monte-Carlo (MCMC) sampling and the best fitting models were selected by comparing the Akaike and Bayesian Information Criteria (AIC and BIC respectively) weights (Fig 2E and 2F, left). These weights represent a normalisation of AIC and BIC values calculated by dividing the AIC and BIC values by the sum of these values across all models (log-likelihood of model given data punished for increasing complexity in two different ways). This is convenient as it allows these values to be transformed into a probability space and hence become comparable across samples [63]. The model with the highest weight explains the data best. For excitatory mossy fiber synaptic transmission, a model containing a single facilitating variable with an exponent of 2 ($f^2$) (Eqs 1 and 2) best explained the experimental data (Figs 2E and S2A, S2C).

$$\text{EPSC amplitude} = g_E^{MAX} f^2 (V_{mem} - E_{GLUT}) \tag{1}$$

$$\frac{df}{dt} = \frac{f_0 - f}{\tau_f} + a(1-f)\sum_s \delta(t - t_s) \tag{2}$$

where $V_{mem}$ is the holding voltage of the cell in voltage clamp, $g_E^{max}$ is the maximum excitatory conductance, $E_{GLUT}$ is the reversal potential of glutamatergic transmission determined in S1 Fig, and $t_s$ is the timing of the $s^{th}$ spike (or pulse). Explanation of other parameters for the short-term plasticity model is given in the Materials and Methods section.

It is noticeable that AIC and BIC weights disagree on which model best explains the data. The $f^2$ model had only the second highest AIC weight, but had the highest BIC weight, whereas the more complex $af$ model had a higher AIC weight. However, the evidence ratio for BIC points favours the $f^2$ model (P($f^2$|Data)/P($af$|Data) = 8.92 (see [64]), whereas the evidence ratio for AIC indicates little evidence in favour of the $af$ model (P($af$|Data)/P($f^2$|Data) = 1.51). Together this demonstrates that the $f^2$ model best explains the data.

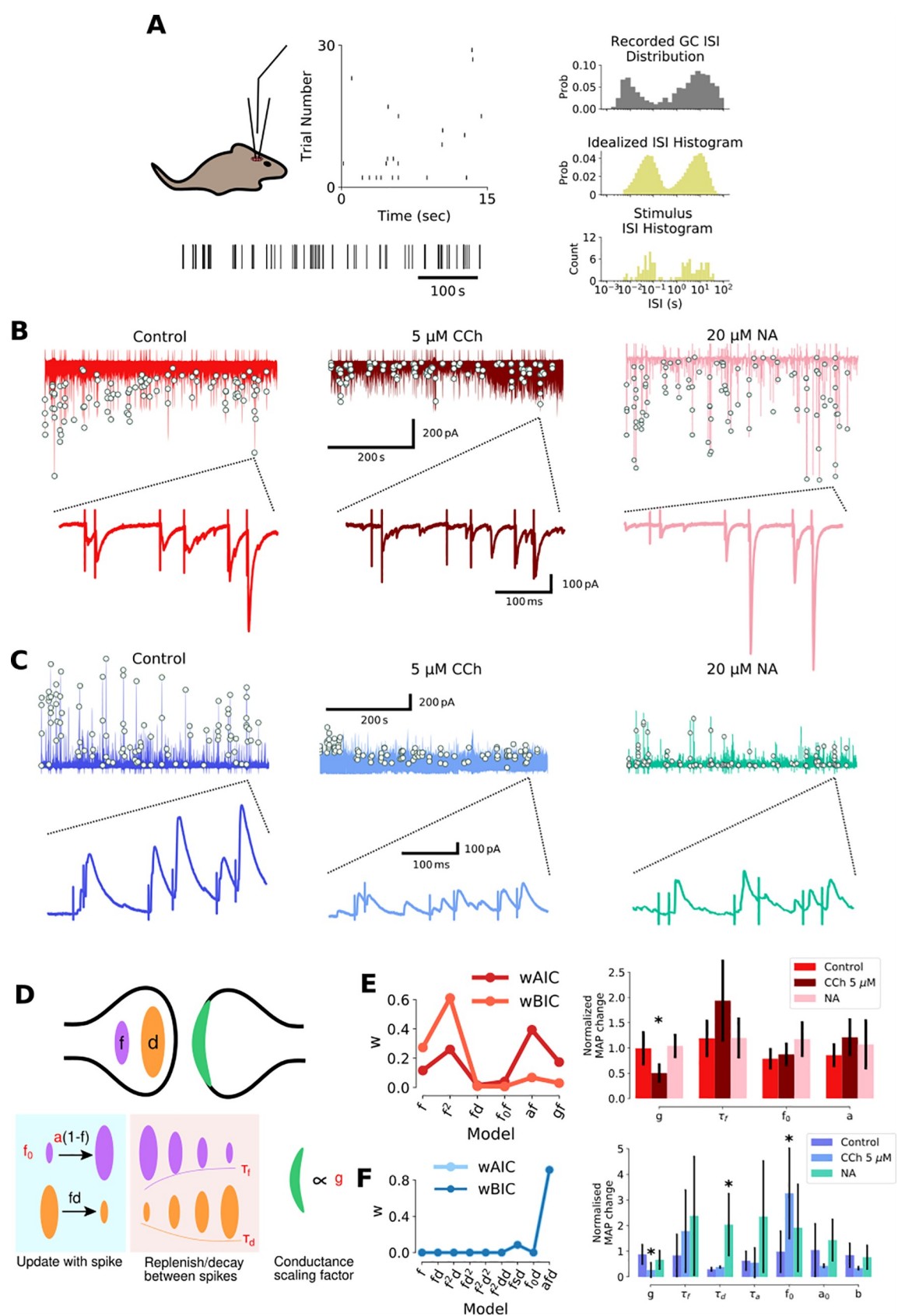

**Fig 2. The mechanism of carbachol and noradrenaline action on excitatory and inhibitory feed-forward mossy fiber transmission determined by short-term plasticity models.** A) Irregular stimulation protocol modelled on naturalistic granule cell spike patterns. GC spike patterns recorded *in vivo* during a spatial memory task, with a bimodal inter-spike interval (ISI) distribution (top right). Bimodal ISI distribution modelled as a doubly stochastic Cox process (middle right), with irregular stimulation protocol a sample drawn from this process (bottom right). B-C) Experimentally recorded EPSCs and IPSCs evoked by irregular stimulation protocol in hippocampal slices under control or presence of 5 µM CCH or 20 µM NA. Evoked peaks highlighted by white dots. The same example burst is shown on expanded timescales. D) Tsodyks-Markram short-term plasticity model schematic illustrating facilitating (*f*) and depressing (*d*) presynaptic components with time constants ($\tau_f$, $\tau_d$) and postsynaptic scaling factor (*g*). E-F) Model selection and fitting for EPSCs (E) and IPSCs (F). Left: AIC and BIC weights for each fitted model. Model selection by highest AIC and BIC weights and evidence ratios. Right: Modulation by CCh or NA assessed by effect on parameter fits normalized by time-matched control. Error bars are standard deviations, * denotes significant parameter change.

For inhibitory feed-forward mossy fiber synaptic transmission, a complex model with facilitation (*f*), depression (*d*), and additional facilitation over the increment parameter *a* (*afd*) produced the best fit (Figs 2F and S2, Eqs 3, 4, 5 and 6) with both AIC and BIC weights convincingly pointing to the *afd* model as most appropriate to describe the data. Additional parameters in this model included a time course for facilitation of *a* ($\tau_a$), an increment scaling factor for *a* (*b*), and baseline ($a_0$).

$$\text{IPSC amplitude} = g_I^{MAX} f\ d(V_{mem} - E_{GABA}) \tag{3}$$

$$\frac{df}{dt} = \frac{1-d}{\tau_d} + f\ d\sum_s \delta(t - t_s) \tag{4}$$

$$\frac{dd}{dt} = \frac{1-d}{\tau_d} + f\ d\sum_s \delta(t - t_s) \tag{5}$$

$$\frac{da}{dt} = \frac{a_0 - a}{\tau_a} + b_a(1 - a)\sum_s \delta(t - t_s) \tag{6}$$

where $g_I^{max}$ is maximum inhibitory conductance and $E_{GABA}$ is the reversal potential of GABAergic transmission determined in S1 Fig. See Materials and Methods for detailed explanation of the above equations.

Discrepancies between samples drawn from posterior-predictive distributions of these models indicated good fit for both models (S2A and S2B Fig).

Using the $f^2$ and *afd* models for the activity-dependent progression of excitatory and inhibitory synaptic weights respectively we were then able to investigate the effect of acetylcholine or noradrenaline on short-term plasticity by comparing normalized parameter estimates to time matched controls. Since posterior distributions for EPSC data were narrow and unimodal, maximum a posteriori (MAP) estimates were used, whereas mean parameter estimates were used for IPSC data since posterior distributions were wide and bimodal in some cases (S2C and S2D Fig). This analysis revealed the small decrease in EPSC amplitude caused by CCh resulted from a reduction in the conductance scaling parameter '*g*' (Fig 2E; 49.9 ± 17.6% (mean ± std)) in agreement with the data in Fig 1B and indicating a postsynaptic mechanism. The substantial decrease in IPSC amplitude caused by CCh resulted from a large reduction in the conductance scaling parameter '*g*', and an increase in the baseline parameter '$f_0$' which also had the effect of reducing facilitation (Fig 2F; 73.1 ± 27.9% decrease in '*g*'; 225.7 ± 160.1% increase in '$f_0$' (mean ± std)). Since IPSCs are disynaptic, it is not straightforward to interpret how these parameter changes reflect biophysical changes to synaptic transmission, but the most likely explanation is a combination of increased feed-forward interneuron excitability

and spike rate, coupled with a strong depression of GABA release. In contrast to acetylcholine, noradrenaline was found to have no effect on EPSC parameters and the reduction in IPSC amplitudes was found to result from an increase in the depression recovery parameter $\tau_d$. Again, it is not straightforward to interpret this change but broadly it represents a slower recovery from depression leading to reductions in synaptic strength upon repeated stimulation across longer timescales.

A certain degree of redundancy exists among the parameters of Tsodyks-Markram type models of short-term plasticity as multiple parameters control the scaling of amplitudes in response to a spike. To explore this, we examined the covariance structure of the posterior distribution over short-term plasticity parameters [65,66] (S3A and S3B Fig). We observed strong correlations amongst the '$g$', '$f_0$', and '$a$' parameters in our excitatory STP model, and strong correlations between the '$g$', '$\tau_d$', '$f_0$', '$a_0$', and '$b$' parameters in the inhibitory STP model. This indicates that there were many parameter sets that could explain our results. However, we chose the priors over these parameters used in our MCMC inference procedure to bias our results towards smaller parameter values, since we reasoned these corresponded to lower energy costs. Indeed, we observed that the mode of joint distributions observed tended to lie in regions of the parameter space that minimized these scaling parameters (S3C and S3D Fig). This indicates any effects observed due to cholinergic or noradrenergic agonists were inducing parameter changes in a regime with realistic constraints.

These results highlight the differential effects of acetylcholine and noradrenaline on mossy fiber synaptic transmission and enable investigation of the granule cell spike patterns that favor excitation over inhibition in the presence of each neuromodulator.

## Enhancement of mossy fiber Excitatory-Inhibitory balance by acetylcholine and noradrenaline

Feed-forward inhibition dominates excitation in the mossy fiber pathway for the majority of spike patterns [49,59]. Since acetylcholine and noradrenaline depress inhibitory transmission more than excitatory transmission (Figs 1 and 2) it is expected that the Excitatory-Inhibitory balance will be shifted towards excitation but the precise spike patterns that this occurs for are unclear. Furthermore, since the mechanisms by which this increase in Excitatory-Inhibitory ratio occur are different for acetylcholine and noradrenaline it is likely that the spike patterns favouring excitation will be different for each neuromodulator. To examine how Excitatory-Inhibitory balance is affected by acetylcholine or noradrenaline with different spike patterns we first tested the dependence of short-term synaptic dynamics on background firing rate using the $f^2$ and $afd$ models for short-term plasticity dynamics of excitatory and inhibitory transmission (Materials and Methods and Table 1). Spike patterns were described by two

**Table 1. Best fit Tsodyks-Markram model parameter sets for EPSC and IPSC short-term plasticity.** Values that are changed by acetylcholine or noradrenaline are shown.

| Parameter | EPSC | IPSC |
|---|---|---|
| $g^{MAX}$ (nS) | 6.6 (3.3 for acetylcholine) | 26.0 (6.7 for acetylcholine) |
| $\tau_f$ (s) | 3.3 | 1.4 |
| $f_0$ | 0.3 | 0.05 (0.16 for acetylcholine) |
| $a$ | 0.15 | |
| $\tau_d$ (s) | | 0.8 (1.6 for noradrenaline) |
| $\tau_a$ (s) | | 8.0 |
| $a_0$ | | 0.08 |
| $b$ | | 0.11 |

parameters: a between burst interval $\Delta t_{between}$ describing a background firing rate, and a within burst interval $\Delta t_{within}$ describing the time between spikes in a burst. The steady state value of $f$, $d$, and $a$ given $\Delta t_{between}$ were then used to replace their baseline values ($a_0 \rightarrow a_\infty$; $f_0 \rightarrow f_\infty$; $d_0 = 1 \rightarrow d_\infty$) to set their initial values at the beginning of a burst, i.e.,

$$a_\infty = \frac{a_0\exp(\Delta t_{between}/\tau_a) - a_0 + b_a}{\exp(\Delta t_{between}/\tau_a) - 1 + b_a} \tag{7}$$

$$f_\infty = \frac{f_0\exp(\Delta t_{between}/\tau_f) - f_0 + a_\infty}{\exp(\Delta t_{between}/\tau_f) - 1 + a_\infty} \tag{8}$$

$$d_\infty = \frac{1 - \exp(-\Delta t_{between}/\tau_d)}{1 + (1 - f_\infty)\exp(-\Delta t_{between}/\tau_d)} \tag{9}$$

By systematically varying the between and within burst intervals for both excitatory and inhibitory synaptic input we were able to simulate EPSCs and IPSCs in the presence and absence of acetylcholine or noradrenaline using parameter values given in Table 1 (Fig 3A). The amplitudes of these responses were then used to explore the effects of acetylcholine or noradrenaline on within burst facilitation ratios and Excitatory-Inhibitory balance across a wide range of between and within burst intervals.

Experimental data shows that mossy fiber EPSCs are exquisitely sensitive to between burst interval with facilitation (also known as frequency facilitation) revealed as between burst interval is decreased. Furthermore, it has been shown that shortening the between burst interval decreases within burst facilitation [53]. Our simulations replicated this interdependence of between and within burst interval with respect to EPSC facilitation with values closely associated with the experimental data in the literature [47,53] (Fig 3B and 3C). Since noradrenaline had no effect on EPSCs and the CCh induced depression of EPSCs was mediated by a reduction in synaptic conductance, neither CCh nor noradrenaline altered either between or within burst synaptic facilitation (Fig 3C).

The situation for inhibitory synaptic transmission was more complex. Over the course of a burst synaptic amplitude facilitation was greatest when between and within burst intervals were largest. As between and within burst intervals reduced, the facilitation morphed into a depression towards the end of the burst resulting in limited inhibition at the end of high frequency bursts (Fig 3B and 3D). CCh depressed the initial IPSC amplitude and dramatically reduced subsequent facilitation within bursts at all between and within burst intervals (Fig 3B and 3D). In contrast, noradrenaline produced much more subtle effects with only small reductions in IPSCs across the range of between and within burst intervals (Fig 3B and 3D).

We then combined the results from excitatory and inhibitory facilitation to estimate EPSC-IPSC amplitude ratios over the course of a burst. In control conditions, excitation dominates over inhibition only after multiple spikes in a burst and when bursts occur at shorter between and within burst intervals (Fig 3E) [36,49,59]. However, in the presence of CCh, excitation dominates over inhibition at earlier stimuli within the burst, and over longer between and within burst intervals meaning cholinergic receptor activation allows excitation to dominate over inhibition for a broader range of stimulus patterns and specifically for physiologically relevant stimulus patterns containing high frequency bursts with long inter-burst intervals (Fig 3E) [52,61,62]. In contrast, noradrenaline only marginally enhanced EPSC-IPSC amplitude ratios across the range of intervals and stimuli (Fig 3E).

We next investigated the biophysical effects of the acetylcholine- or noradrenaline-induced reduction in feed-forward inhibitory synaptic transmission at mossy fiber synapses. In

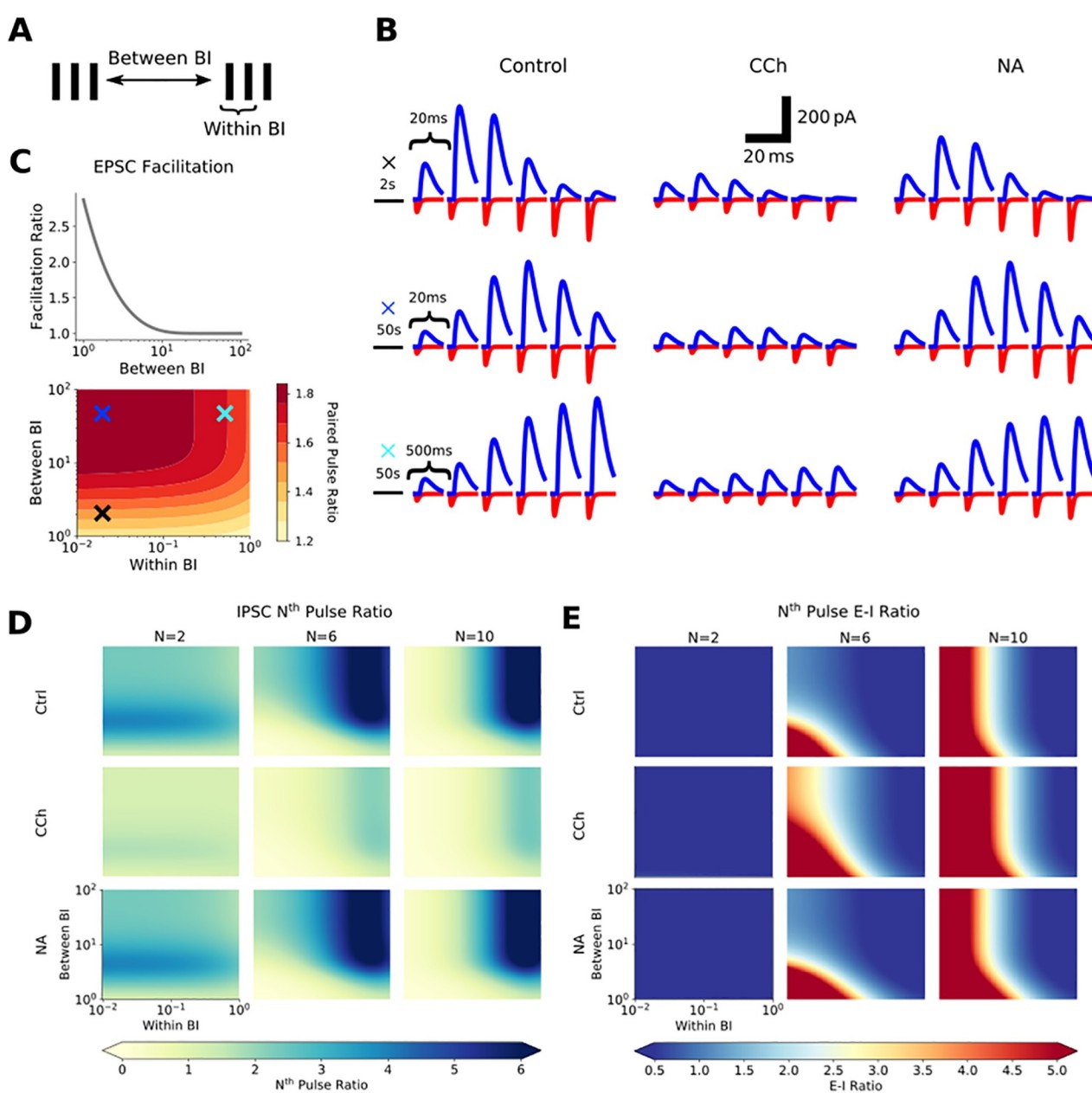

**Fig 3. Carbachol and noradrenaline alter the Excitatory-Inhibitory ratio within the feed-forward mossy fiber pathway in a frequency-dependent manner.** A) Simplification of bursting spike trains into two parameter spaces: between burst interval (BI) describing interval between bursts in a spike train, and within BI describing interval between spikes within a burst. B) Example synaptic waveforms of expected mossy fiber EPSCs (red) and IPSCs (blue) generated by the short-term plasticity models under control, CCh and NA conditions. The 3 rows illustrate short-term plasticity dynamics at three pairs of within and between burst intervals (20ms and 2s, 20ms and 50s, 500ms and 50s respectively). C) Expected short-term plasticity of EPSCs across a wide range of within and between BIs. Light blue, dark blue and black crosses shown in C denote within and between BIs used in the examples shown in B. Data in the presence of CCh or NA not shown since CCh does not change the facilitation of EPSCs and NA has no effect on EPSCs. D) Expected short-term plasticity of IPSCs across a wide range of within and between BIs and in control, CCh and NA conditions. Pulse ratios for 2nd, 6th and 10th pulses compared to the 1st are shown to illustrate change in facilitation across a 10 pulse burst. E) Progression of Excitatory-Inhibitory ratio across the range of within and between BIs and in control, CCh and NA conditions. Pulse ratios for 2nd, 6th and 10th pulses compared to the 1st are shown to illustrate change in E-I ratio across a 10 pulse burst.

particular, the modulation of back-propagating action potentials and EPSPs in CA3 pyramidal cells that are critical for the induction of long-term potentiation (LTP) at recurrent CA3-CA3 synapses and therefore the formation of CA3 ensembles [41–43]. Mossy fibers provide powerful excitatory drive to the soma of CA3 pyramidal cells and have been referred to as 'conditional detonator' synapses because a single synapse can trigger postsynaptic action potentials in response to high frequency bursts of presynaptic action potentials but not single action potentials [37–39]. We hypothesized that feed-forward inhibitory synaptic transmission reduces back-propagating action potentials and mossy fiber evoked EPSPs which will inhibit or prevent the induction of LTP [67–69], and that acetylcholine, and possibly noradrenaline, will relieve this inhibition by reducing feed-forward inhibition. To test this, we used a well characterized multi-compartment biophysical model of a CA3 pyramidal cell that recapitulates key physiological attributes of CA3 firing patterns [70,71] with 15 different reconstructed morphologies selected from Neuromorpho.org [72,73]. Our model incorporated mossy fiber excitatory synaptic input on the very proximal portion of the apical dendrite where conductance was set for each cell morphology to induce spiking after the 4th input spike in the absence of inhibition, as demonstrated in slice recordings [74]. We also included a single average feed-forward inhibitory synapse per dendritic compartment within 400μm of the somatic compartment, corresponding to dendritic inhibition targeting the stratum lucidum and stratum radiatum (Fig 4A) [45]. Total inhibitory conductance was set in relation to the tuned excitatory conductance, and divided equally among inhibitory synapses [45]. Membrane potential and the resultant intracellular calcium concentration were simulated across multiple somatic and dendritic compartments of a reconstructed CA3 pyramidal cell incorporating the thin oblique dendrites in stratum radiatum where the majority of CA3-CA3 recurrent synapses are located [75]. With feed-forward inhibition intact, action potentials (defined as crossing a threshold of 0 mV) and EPSPs back-propagate into the principal dendritic shafts without much change in amplitude but are rapidly attenuated on entering the thin oblique dendrites (Fig 4B and 4C). This leads to minimal calcium influx through voltage-gated calcium channels at these dendritic sites (Fig 4B). However, when feed-forward inhibition is reduced by acetylcholine, attenuation of action potentials and EPSPs is greatly reduced allowing substantial calcium influx (Fig 4B and 4C). The relief of action potential and EPSP attenuation by acetylcholine was selective for the oblique dendrites in stratum radiatum, was consistent for multiple different CA3 pyramidal cell morphologies (Figs 4C and S4), increased both the amplitude and probability of action potential back-propagation (Fig 4E and 4G) and reduced the number of stimuli in a train required to trigger back-propagating action potentials and substantial dendritic calcium transients (Fig 4D). In contrast, the more marginal reduction in feed-forward inhibition caused by noradrenaline meant that although the probability that action potentials back-propagated into dendrites increased with noradrenaline, the number of action potentials initiated during a train of stimuli did not increase (Fig 4D and 4E). We also examined how the probability of back-propagating action potentials reaching oblique dendrites was affected by variations in the Excitation-Inhibition ratio across a range of values (Fig 4H). Keeping the excitatory mossy fiber input conductance fixed, control conditions corresponded to gI/gE = 3 and cholinergic conditions to gI/gE = 1. Increasing disinhibition strongly increased the probability of back-propagating action potentials reaching oblique dendrites, with noradrenaline lying between gI/gE = 1 and 3. Due to the considerable variability in mossy fiber responses and the steeper increase in action potential back-propagation probability around gI/gE = 1, the effect of acetylcholine on $Ca^{2+}$ influx at oblique dendrites is likely to be greater than modelled here, whereas the effect of noradrenaline is likely to be well estimated due to the even gradient between gI/gE = 1 and 3.

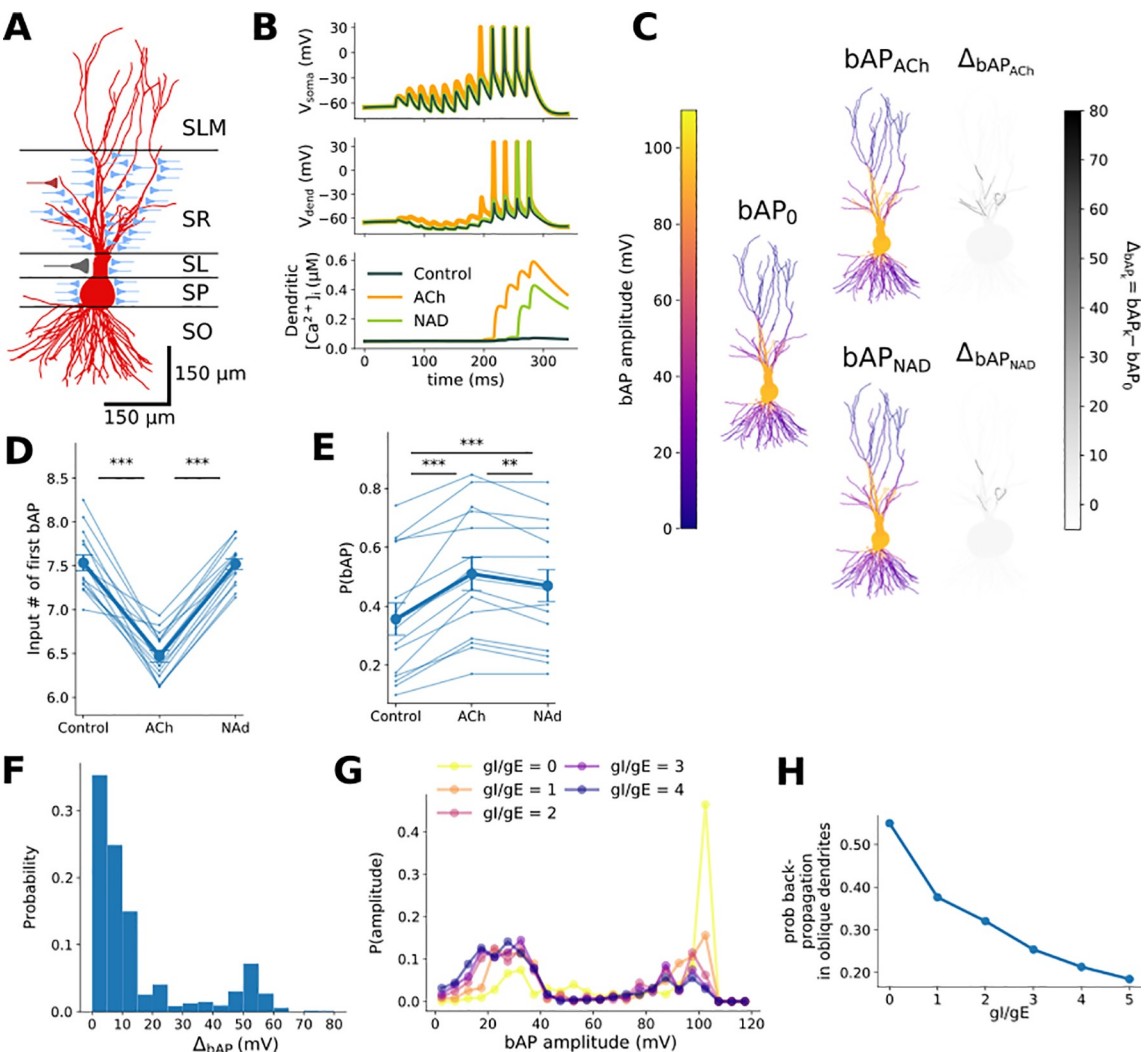

**Fig 4. Acetylcholine- and noradrenaline-mediated disinhibition facilitates back-propagation of EPSPs and action potentials into the dendrites of CA3 pyramidal cells.** A) Sketch of CA3 pyramidal cell and positioning within the layers of the hippocampus. Synaptic inputs are shown with location of contact (red—recurrent CA3-CA3 synapse, blue—feed-forward inhibitory synapse, gray—mossy fiber synapse). SLM–stratum lacunosum moleculare, SR–stratum radiatum, SL–stratum lucidum, SP–stratum pyramidale, SO–stratum oriens. B) Example traces produced by the biophysical CA3 neuron model of action potentials generated at the soma from summated mossy fiber EPSPs presented at 20 Hz (top), back-propagation into the radial oblique dendrites (middle), and dendritic calcium influx (bottom), in control and with acetylcholine (ACh) or noradrenaline (NA)-mediated disinhibition of feed-forward inhibition. C) Back-propagating action potential amplitude before (left) and after (middle) cholinergic or noradrenergic modulation, and the difference in amplitude (right) distributed across an example CA3 pyramidal cell. D) The number of stimuli required to generate a back-propagating action potential across all cell morphologies. Only dendrites that had back-propagating action potentials in control conditions are shown. E) The proportion of oblique dendrites in stratum radiatum reached by a back-propagating action potential per stimulus for all cell morphologies. F) Histogram of differences in back-propagating action potential amplitudes with and without acetylcholine disinhibition in stratum radiatum oblique dendritic compartments (< 1 μm diameter) from 15 cells. G) Distribution of back-propagating action potential amplitudes in stratum radiatum oblique dendrites for a range of excitation-inhibition ratios in 15 cells. In our simulations the effect of acetylcholine was modelled as a change in this ratio (the absence of acetylcholine, gI/gE = 3; in the presence of acetylcholine, gI/gE = 1). H) The probability of successful action potential back-propagation (bAP amplitude > 40 mV) in oblique dendrites in 15 cells as a function of excitation-inhibition ratio.

The different mechanisms and extent of mossy fiber feed-forward disinhibition by acetylcholine or noradrenaline has major implications for synaptic plasticity at CA3-CA3 recurrent synapses, since spike timing-dependent plasticity is dependent on the back-propagation of action potentials and EPSPs, postsynaptic calcium accumulation and activation of calcium-

dependent signalling pathways [41,43]. Our simulations indicate that disinhibition of the feed-forward mossy fiber pathway by acetylcholine or noradrenaline are important to facilitate the induction of synaptic plasticity between CA3 pyramidal cells when CA3 ensembles are activated by mossy fiber inputs at physiologically relevant patterns. Furthermore, acetylcholine reduces the number of stimuli within a burst required to generate substantial dendritic $Ca^{2+}$ whereas noradrenaline only facilitated the back-propagation of action potentials once they were initiated and is therefore only important for $Ca^{2+}$ signalling driven by longer bursts with more stimuli. This predicts that the different mechanisms by which acetylcholine and noradrenaline enhance Excitatory-Inhibitory ratio translate into selective enhancement of dendritic calcium signalling in response to specific patterns of feed-forward mossy fiber synaptic transmission.

## Ensemble formation in CA3 driven by mossy fiber input

To further investigate the effects of acetylcholine or noradrenaline on the creation of CA3 ensembles by mossy fiber input we next turned to a spiking network model of CA3. This network was comprised of point neurons with Izhikevich-type dynamics [76] parameterized to reproduce spiking patterns for excitatory CA3 pyramidal cells and inhibitory fast spiking interneurons [27,77,78] connected in an all-to-all fashion. Subsets of pyramidal cells were driven by excitatory mossy fiber input with short-term facilitation dictated by the model determined in Fig 2. CA3-CA3 recurrent synaptic connections were subject to an experimentally determined symmetric spike timing-dependent plasticity (STDP) rule (Fig 5A) with no short-term plasticity [41]. Since this symmetric STDP rule is inherently unstable (every pre-post spike pair increases synaptic strength), we posit that a homeostatic mechanism tied to postsynaptic activity exists to stabilize network activity during plasticity. As a result we implemented a modification to the symmetric STDP rule to allow for reductions in synaptic strength dependent on the

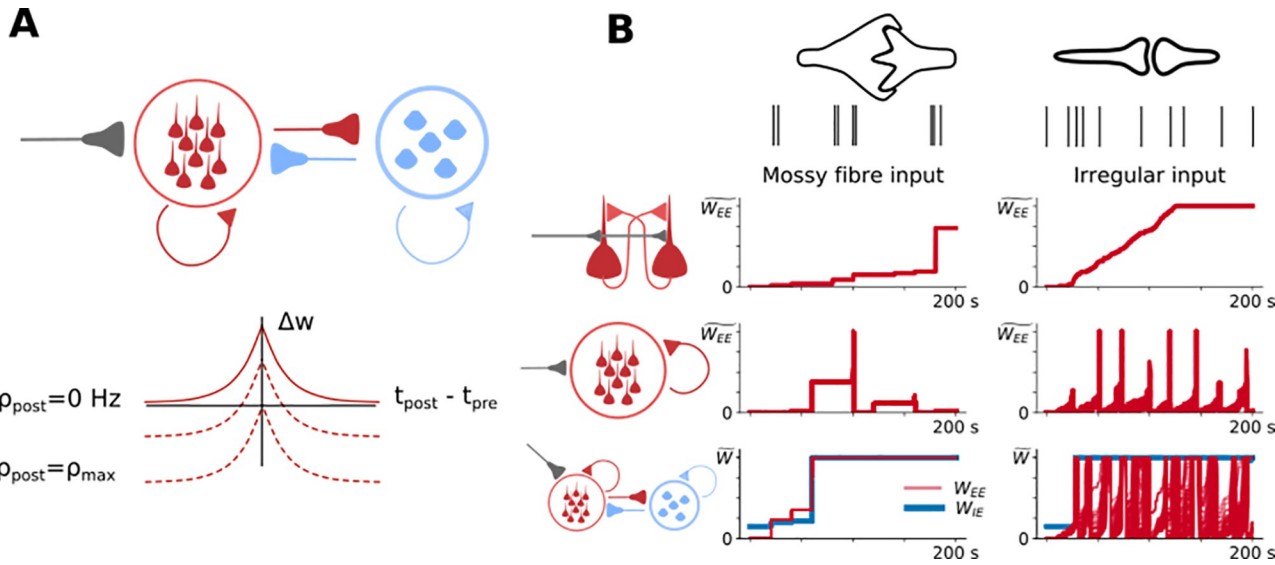

**Fig 5. Facilitating mossy fiber inputs generate rapid and stable ensemble formation.** A) Schematic showing the network properties of the spiking network model and long-term plasticity rules. Left: A population of excitatory (red) and inhibitory (blue) cells with all-to-all connectivity and mossy fiber input (gray). Right: Recurrent excitatory CA3-CA3 spike timing-dependent plasticity rule with a symmetric window that shifted from potentiation for correlated spiking at low rates (Ppost = 0), to depression for uncorrelated spiking near the maximum postsynaptic firing rate (Ppost = Pmax). B) Comparison of mossy fiber to irregular 'perforant path' input. Synaptic weight evolution with time for CA3-CA3 recurrent connections (red) and inhibitory to excitatory connections (blue) for a two cell excitatory population (top), a 10 cell excitatory population (middle) and a population including 10 excitatory and 5 inhibitory cells (bottom).

postsynaptic firing rate through synaptic scaling [79]. At low firing rates, potentiation is induced with small differences in pre- and post-synaptic spike times. As the postsynaptic firing rates increase, large differences in pre- and post-synaptic spike times cause depression. At a maximum firing rate, no potentiation is possible.

Within this network we first characterized the speed and stability of ensemble creation where an ensemble was defined as being formed when all synapses between cells within the same ensemble had reached their maximum weight, and all synapses between cells not within the same ensemble had decreased to zero. In addition, the properties of mossy fiber input were studied in comparison with a more generic input reminiscent of perforant path activity during direct information transfer between entorhinal cortex and CA3 to see how they compared in their ability to drive ensemble formation via synchronous spiking in a small population of cells (Fig 5B). Mossy fiber spike patterns were modelled as a Poisson process with brief (200 ms every 20 seconds) high intensity (50 Hz) firing rates on a very low basal firing rate (0.2 Hz), and were connected to pyramidal cells by a strong facilitating synapse (3.0 nS). The conductance of the mossy-fiber input was chosen to elicit a 1mV EPSP amplitude from a single spike [59]. This firing pattern represents a strongly separated, sparse firing pattern in a single presynaptic cell, which is expected in dentate gyrus granule cells. Perforant path spike patterns were modelled as a population of presynaptic entorhinal cells in the synchronous irregular state modelled as 120 homogeneous Poisson processes firing at 10 Hz with a correlation coefficient of 0.9 and static, weak synapses fixed at 0.1 nS, which broadly reflects entorhinal activity in a freely behaving rat [80–82].

We initially built up the network model sequentially to investigate which components were necessary for the speed and stability of ensemble creation. At first, two CA3 pyramidal cells were connected and driven with only excitatory input. For both mossy fiber and perforant path inputs the cells quickly became connected (Fig 5B, top row). Increasing ensemble size to 10 excitatory cells destabilized the ensemble formation process (Fig 5B, middle row). The destabilization resulted from unbalanced potentiation of recurrent excitation that caused a large increase in the firing rate leading to strong depression or 'resetting' of the synaptic weight with further spiking as a result of synaptic scaling. The addition of 5 feedback inhibitory cells stabilized ensemble formation in the case of mossy fiber input, but not for perforant path input (Fig 5B, bottom row). This was because the perforant path input provided colored noise input amplified by recurrent excitation that caused excitatory cells to fire at high rates too often and feedback inhibition was insufficient to counter this amplification. Mossy fiber input is driven only briefly at sparse intervals, meaning there was little opportunity to exceed target firing rate, and when there was, feedback inhibition was sufficient to contain it. These results show that in this model mossy fiber-like sparse inputs and feedback inhibition within the CA3 recurrent network are important for rapid and stable formation of CA3 ensembles.

## CA3 recurrent synapses and cellular excitability are regulated by acetylcholine but not noradrenaline

The creation of CA3 ensembles depends not only on mossy fiber input from the dentate gyrus but also on the state of the CA3 network. Two critical factors are the strength of CA3-CA3 recurrent synaptic inputs and the excitability of CA3 pyramidal neurons [51] which are both known to be sensitive to acetylcholine [28,57,58] but the action of noradrenaline on these network properties remains largely unknown. Therefore, we next experimentally compared the effects of CCh and noradrenaline on recurrent excitatory CA3-CA3 synaptic transmission and CA3 pyramidal neuron intrinsic excitability. 5 μM CCh caused a ~50% depression in excitatory synaptic transmission (Fig 6A and 6C; 42.4 ± 8.2% of baseline measured at 1st pulse,

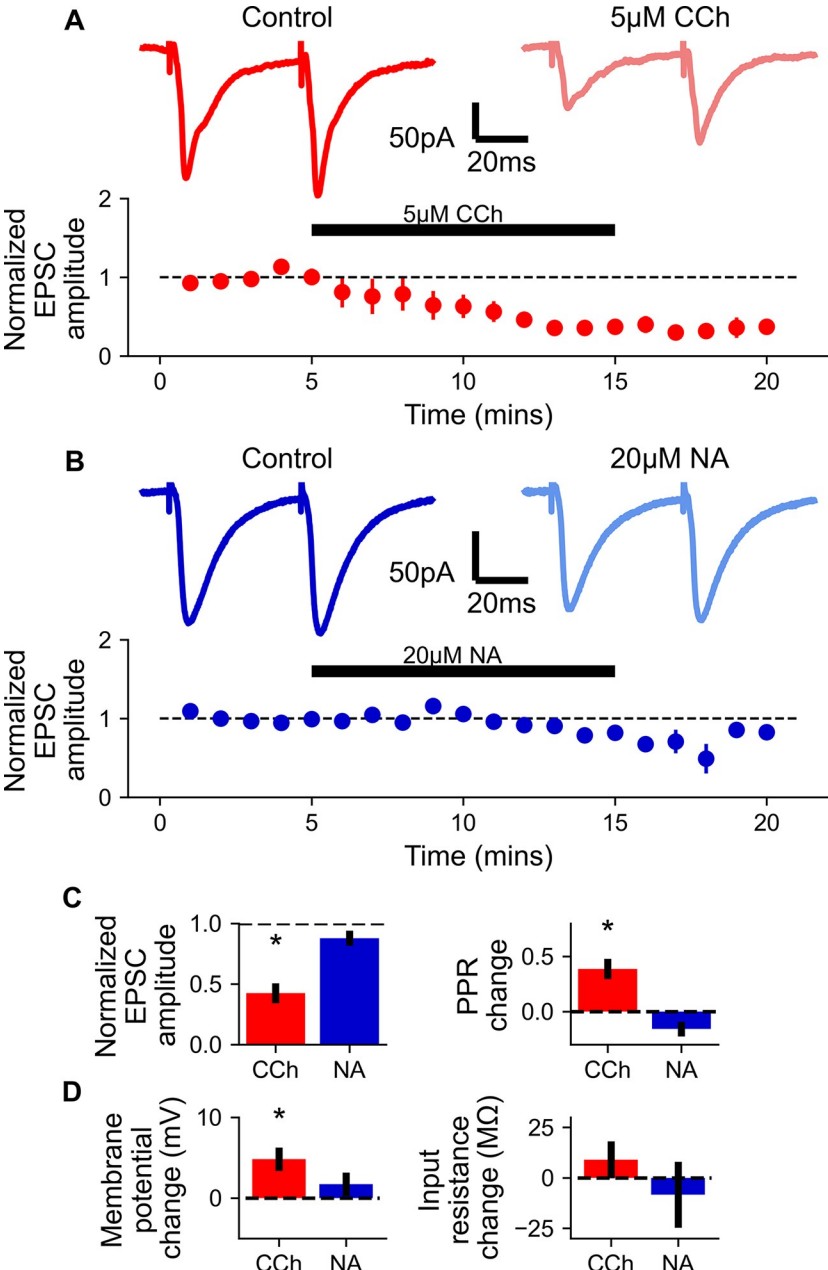

**Fig 6. Carbachol but not noradrenaline enhances CA3 cellular excitability and depresses associational/commissural synaptic connections.** A) 5 μM CCh depresses associational/commissural evoked EPSCs. Top: Example traces before and after bath application of 5 μM CCh. Bottom: Time course of CCh effect, and washout (n = 4). B) 20 μM NA has no effect on associational/commissural evoked EPSCs. Top: Example traces before and after bath application of 20 μM NA. Bottom: Time course of NA effect, and washout (n = 5). C) 5 μM CCh but not 20 μM NA depresses associational/commissural evoked EPSCs and increases paired pulse ratio (PPR). D) 5 μM CCh but not 20 μM NA depolarizes CA3 pyramidal cell membrane potential and increases input resistance.

n = 4, $p < 0.05$) accompanied by an increase in PPR (Fig 6C; $p < 0.05$) in line with previous reports for the action of acetylcholine on neurotransmitter release [28,57,58]. In contrast 20 μM noradrenaline had no effect on excitatory synaptic transmission or PPR (Fig 6B and 6C; 87.8 ± 6.2% of baseline measured at 1st pulse, n = 5, $p = 0.12$; PPR $p = 0.08$). The depolarising

effects of acetylcholine mediated by increases in input resistance are well documented in many neuronal subtypes including CA3 pyramidal cells but the effects of noradrenaline are not known [57,58]. We confirmed that acetylcholine depolarises CA3 pyramidal neurons where CCh caused an ~5mV depolarisation with an associated increase in input resistance (Fig 6D; membrane potential increased by 4.8 ± 1.5 mV, n = 5, $p < 0.05$; Input resistance increased by 9.0 ± 9.2 MΩ, n = 5, $p = 0.24$) but noradrenaline had no effect on membrane potential or input resistance (Fig 6D; membrane potential 1.7 ± 1.4 mV, n = 7, $p = 0.28$; Input resistance -8.3 ± 16.3 MΩ, n = 6, $p = 0.63$). This was supported by the observation that CCh produced an increase in spontaneous synaptic transmission (S1E and S1F Fig). Thus, acetylcholine, but not noradrenaline, has a regulatory role in these CA3 network properties that play an important role in determining the formation of CA3 neuronal ensembles.

## Acetylcholine facilitates mossy-fiber driven ensemble formation in CA3

We next examined how acetylcholine or noradrenaline affects the CA3 network's ability to form ensembles. To achieve this, we implemented the experimentally determined effects of CCh and noradrenaline on CA3 network properties within the spiking network model (see Table 2). Whilst acetylcholine had a somewhat greater facilitatory effect on back-propagating action potentials than noradrenaline (Fig 4), conservatively, we assumed synaptic plasticity at CA3-CA3 recurrent synapses can occur in the presence of either neuromodulator within the spiking network model and not in their absence. In the absence of plasticity no ensemble formation can occur so we only tested ensemble formation in the presence of noradrenaline or acetylcholine. To mimic the effects of CCh the resting membrane potential for excitatory cells was depolarized by 5 mV to -70 mV and inhibitory cells were also depolarized to -63 mV [21,27,83]. Separately, the CA3-CA3 excitatory synaptic conductance was halved (Fig 6A) [27,84]. Noradrenaline was assumed to enable synaptic plasticity between CA3 pyramidal cells and therefore ensemble formation but did not otherwise affect CA3 network properties. Networks contained 64 excitatory and 16 inhibitory cells with excitatory cells grouped into 8 ensembles consisting of 8 cells each (Fig 7A). Network sizes were chosen to reflect ratios of excitatory to inhibitory cells observed in CA3. For computational efficiency we chose to examine small network sizes with all-to-all connectivity (as in [27,85–87]), rather than the more biologically realistic sparse connectivity in larger network sizes [42] studied elsewhere [88–90]. Each cell within a single ensemble received the same excitatory mossy fiber input, which followed a Poisson process with a low firing rate of 0.2 Hz punctuated by bursts for 250 ms every 20 s at varying frequency and short-term plasticity dynamics dictated by the model determined in Fig 2. No two ensembles received bursts at the same time and spike timing-dependent plasticity rules were implemented regardless of alterations in CA3 network parameters representing the presence of acetylcholine. Networks were simulated over 400 seconds.

In the presence of noradrenaline and therefore with synaptic plasticity engaged but without the CA3 network effects of acetylcholine, bursts at a frequency of 30 Hz formed discrete

**Table 2. CA3 network parameter changes to model cholinergic modulation (ACh) of CA3.** Note that noradrenaline (NA) did not cause any changes to CA3 network properties and therefore parameters for noradrenaline are the same as control conditions.

| Parameter | Type | Control & NA | ACh |
| --- | --- | --- | --- |
| $c$ | E cell | -63 mV | -61 mV |
| $d$ | E cell | 60 pA | 50 pA |
| $v_{rest}$ | E cell | -75 mV | -70 mV |
| $v_{rest}$ | I cell | -65 mV | -63 mV |
| $g_{syn}^{-}$ | EE syn | 0.5 nS | 0.25 nS |

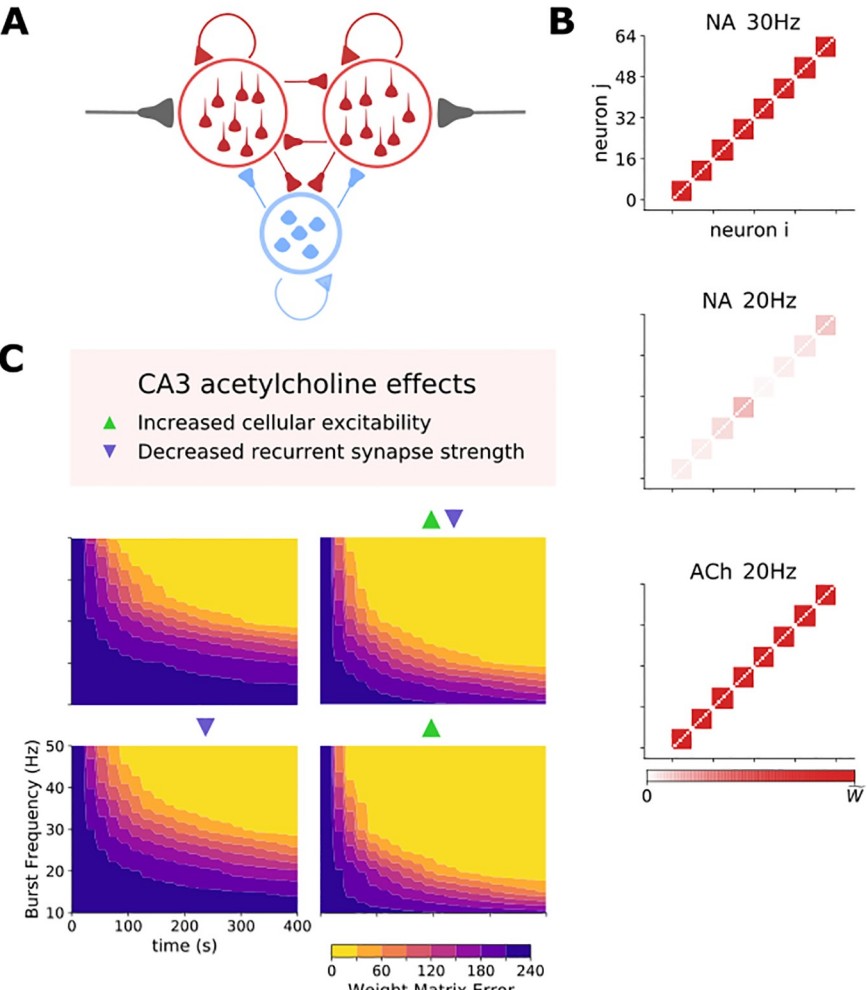

**Fig 7. Acetylcholine speeds up ensemble formation and lowers input frequency requirement by increasing cellular excitability.** A) Network setup: A population of excitatory and inhibitory cells connected in all-to-all fashion. Subpopulations of excitatory cells receive independent feed-forward input that drives ensemble formation. B) Example weight matrix driven by input with 30 Hz bursts in the presence of NA, 20 Hz bursts in the presence of NA or 20 Hz bursts in the presence of acetylcholine. Stronger weights indicate robust ensemble formation. C) Evolution of ensemble formation illustrated by the weight matrix error reduction over time for different input burst frequencies. Triangles denote which effects of acetylcholine on the CA3 network were included in each set of simulations.

ensembles, but these were almost completely abolished when burst frequency was reduced to 20 Hz (Figs 7B and S6A and S6B). Remarkably, when noradrenaline was switched for acetylcholine by applying the observed effects of acetylcholine on the CA3 network, ensemble formation was rescued at the lower burst frequency (Figs 7B and S6C). To quantify network ensemble formation performance and the impact of acetylcholine in greater detail, we used an error metric ($WME$) defined as the summed absolute difference between target and actual weight matrix ($W_{ij}^{actual}$), where the target weight matrix ($W_{ij}^{target}$) is maximum synaptic weights between cells $i$ and $j$ in the same ensemble, and zero weight otherwise, i.e.,

$$WME = \sum_{ij} |W_{ij}^{actual} - W_{ij}^{target}| \tag{10}$$

This analysis revealed that acetylcholine lowers the required input frequency and increases the speed at which ensembles form (Fig 7C). To test which effects of acetylcholine were critical for these aspects of ensemble formation we removed each parameter change in turn. This revealed that the key factor was the increase in cellular excitability, as removing the parameter changes to cellular excitability abolished the effect of acetylcholine but removing reductions in CA3-CA3 recurrent connections did not (Figs 7C and S5A).

Within the CA3 network multiple often highly overlapping ensembles may be encoded. Theoretically this increases the capacity of information encoding but reduces the fidelity of retrieval with a necessary trade-off between these two parameters. This may support generalization, in that it provides a mechanism to associates discrete events spread disparately in time [91]. Therefore, we compared the impact of acetylcholine and noradrenaline on the ability of the CA3 network to reliably encode overlapping ensembles. To incorporate overlap between ensembles the total network size was made variable whilst still containing 8 ensembles of 8 cells each and overlap was introduced by having a subset of excitatory cells receive input from two sources rather than one. Overlap was arranged in a ringed fashion such that each ensemble shared a certain number of cells with their adjacent 'neighbour' (Fig 8A). Retrieval was studied by comparing the population rates of each ensemble, with a smaller difference in rates meaning lower discrimination and more difficult retrieval.

In the presence of noradrenaline, stable ensembles could be formed when the degree of overlap was low but discrete ensembles could not be formed with levels of overlap >1 (Figs 8B and 8C and S7A). In contrast, in the presence of acetylcholine the network could safely support an overlap of 3 cells between discrete ensembles (Figs 8B and 8C and S7B). To test which effects of acetylcholine mediated the enhanced discrimination between overlapping ensembles we removed each parameter change in turn. This revealed that the key factor was the reduction in CA3-CA3 recurrent synapse efficacy since removing this parameter abolished the ability for acetylcholine to allow greater ensemble overlap. In contrast, removing the increase in cellular excitability only increased the time taken to form stable ensembles without affecting the final degree of overlap supported (Figs 8C and S5B). Interestingly, the increase in stable ensembles with significant overlap was associated with a decrease in the discrimination between ensembles during retrieval of information as more overlap produced less separation of population rates between ensembles (Fig 8D). Taken together these data indicate that acetylcholine increases the number of discrete ensembles that may be contained within a finite CA3 network compared to noradrenaline but that this comes at a cost of reduced retrieval fidelity. This implicates a possible role for acetylcholine promoting formation of generalizable ensemble while noradrenaline promotes discriminative ensemble formation.

## Discussion

In this study, we investigated the effects of acetylcholine and noradrenaline on the ability of mossy fiber input from dentate granule cells to form ensembles within the CA3 recurrent network. Experimentally, we discovered that acetylcholine dramatically reduces feed-forward inhibition in the mossy fiber pathway whilst having limited effect on mossy fiber excitatory transmission whereas noradrenaline causes a smaller reduction in feed-forward inhibition with no effect on excitatory transmission (Figs 1 and 2). This disinhibition causes a positive shift in the Excitatory-Inhibitory balance for both neuromodulators which is more pronounced for acetylcholine (Fig 3) creating the conditions required for LTP at recurrent CA3-CA3 synapses and therefore ensemble formation (Fig 4). Most strikingly, acetylcholine, but not noradrenaline, altered CA3 network properties by depolarizing CA3 pyramidal cells and reducing the strength of recurrent CA3-CA3 connections which has important

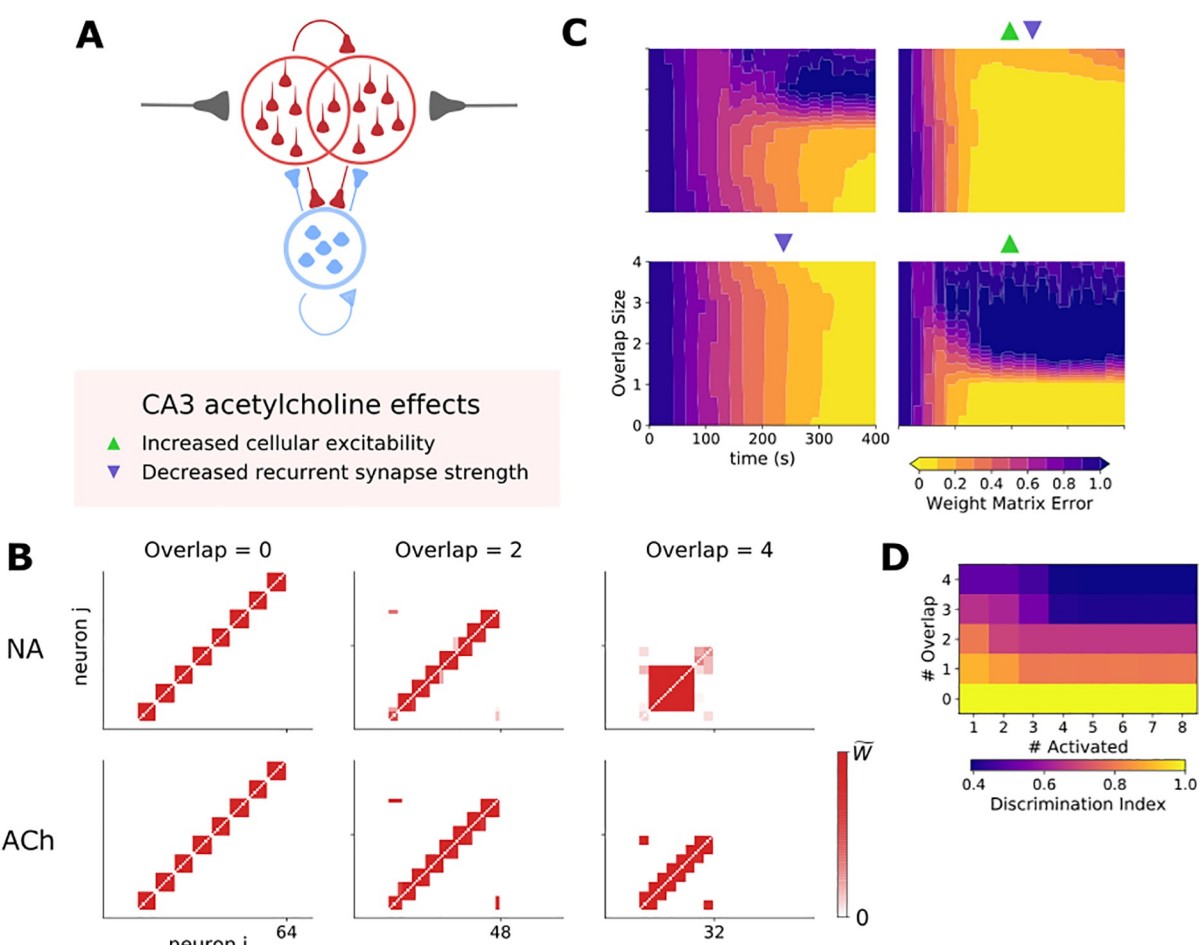

**Fig 8. Acetylcholine enables a CA3 network to form stable overlapping ensembles by reducing the strength of recurrent excitatory CA3-CA3 synapses.** A) Network setup: A population of excitatory and inhibitory cells connected in all-to-all fashion. Subpopulations of excitatory cells receive overlapping feed-forward input that drives ensemble formation. B) Example weight matrix driven by input with increasing degrees of overlap between ensembles [0, 2, 4 cells] in the presence of NA or ACh. C) Evolution of ensemble formation illustrated by the weight matrix error over time for different degrees of overlap between ensembles. Triangles denote which effects of acetylcholine were included in the simulation. D) Effect of increasing overlap between ensembles on the ability to discriminate between ensembles defined as the difference in ensemble population spiking rates.

implications for the formation of ensembles within the CA3 network. We demonstrate this in a computational spiking network model where the separate effects of acetylcholine on cellular excitability and basal CA3-CA3 synaptic strength were found respectively to enhance the robustness of ensemble formation (Fig 7) and the amount of allowable overlap between ensembles (Fig 8). Together, these findings indicate separate mechanisms and roles for acetylcholine and noradrenaline in gating and facilitating memory formation in the CA3 network. Each neuromodulator represents an important salience cue to signal when new ensembles may be formed and therefore which information to encode but those formed in the presence of acetylcholine have higher densities and are more robust than those formed in the presence of noradrenaline.

The three separate mechanisms we identify as important for ensemble formation are likely supported by different cholinergic and noradrenergic receptors at distinct cellular and subcellular locations. In contrast to the presynaptic actions of cholinergic receptors at other hippocampal synapses [58], our experimental data show only a small effect of cholinergic agonists at

excitatory mossy fiber synapses onto CA3 pyramidal cells indicating a limited direct synaptic modulation by muscarinic or nicotinic receptors mediated by postsynaptic changes [56,57,92]. The lack of presynaptic changes indicates no indirect mechanism via enhancement of interneuron spiking and activation of presynaptic GABAB receptors by GABA spillover [57,93]. For similar reasons, our data do not support a role for presynaptic nicotinic receptors ([94] but see [57]) or an increase in dentate granule cell spike frequency [57] since both would be expected to cause an increase in EPSC amplitude by presynaptic mechanisms. The large depression in feed-forward inhibitory mossy fiber transmission by cholinergic activity most likely results from a combination of an increase in feed-forward interneuron excitability and spike rate, mediated by a combination of muscarinic M1 and M3 receptors and nicotinic receptors [57,58,95], coupled with a strong depression of GABA release, mediated by presynaptic M2 receptors present on interneuron terminals [96]. This potentially accounts for the observed increase in basal synaptic release to initial stimulation but overall large reduction in synaptic conductance and release over the course of a burst of stimuli seen in our experimental data and short-term plasticity model. In contrast, the mechanism of action for noradrenergic depression of feed-forward inhibitory mossy fiber transmission is largely unknown but likely mediated by β-adrenergic receptors that slow the recovery from depression [30,97]. Finally, the depression in basal CA3-CA3 recurrent synaptic strength is reported to result from the activation of presynaptic M4 receptors [58]. These mechanistic conclusions predict that M2 muscarinic receptors on interneuron terminals or β-adrenergic receptors are important for disinhibition of mossy fiber feed-forward inhibition necessary for ensemble formation, M4 muscarinic receptors on CA3-CA3 recurrent axon terminals are important for increasing the amount of permissible overlap between ensembles and M1 muscarinic receptors on CA3 pyramidal cells facilitate the rapid and stable formation of ensembles.

The observed differences between exogenous applications of acetylcholine and noradrenaline will likely be accentuated under endogenous neuromodulator release. The concentration of CCh used (5 μM) produces effects in the hippocampus that closely match those caused by optogenetically induced release so endogenous acetylcholine release would likely have similar effects to those we report here [98] and the timecourse of cholinergic receptor activation produced by bath application of CCh is a reasonable approximation of tonic levels of acetylcholine release *in vivo* [15,99]. The concentration of noradrenaline used (20 μM) is somewhat higher than that thought to be evoked by endogenous release in CA1 [31,100] but there is a higher density of noradrenergic fibers in CA3 than CA1 and more direct synaptic targeting [16,101,102] making it likely that the effective concentration of noradrenaline in CA3 is likely in the micromolar range but lower than 20 μM. Therefore, the effects of noradrenaline on feed-forward inhibition that we report may be greater than those expected by endogenous release of noradrenaline. This means it is possible that synaptic noradrenaline release has marginal effects on CA3 network processing although unlikely given the importance of β-adrenergic receptor signalling for CA3-dependent memory [30]. It is also possible that dopamine release from locus coeruleus fibers in the hippocampus is more important than noradrenaline for the formation of CA3 ensembles [16,102]. In addition, the specific effects of acetylcholine and noradrenaline on mossy fiber transmission, CA3 excitability and CA3-CA3 synaptic transmission are likely not the only nodes within the network where these neuromodulators exert actions and the impact of additional network reconfigurations on ensemble formation remain to be explored.

Feed-forward inhibition dominates excitatory transmission in the mossy fiber pathway but unlike other examples of feed-forward inhibition, such as that occurring in the neocortex or CA1 region of the hippocampus, interneurons engaged with mossy fiber feed-forward inhibition target the dendritic compartments of CA3 pyramidal cells as much if not more than the perisomatic areas [45,47,49,103]. This means that mossy fiber feed-forward inhibition strongly

inhibits recurrent CA3-CA3 inputs in stratum radiatum rather than excitatory mossy fiber inputs [104,105]. Our simulations indicate that inhibitory input to dendritic domains in stratum radiatum strongly attenuates the back-propagation of action potentials and EPSPs originating from the somatic compartment into the thin radial oblique dendrites where most of the recurrent CA3-CA3 synapses occur (Fig 4). Since these back-propagating signals are necessary for the induction of LTP at recurrent CA3-CA3 synapses [41,43], this indicates that mossy fiber feed-forward inhibition is well placed to control the induction of LTP at these synapses. Indeed, the highly non-linear nature of dendritic spikes thought to be important for associative LTP at CA3-CA3 recurrent synapses [106] suggests the regulation of this feed-forward inhibition plays a more dominant role in gating LTP than shown in our modelling. The dominance of mossy fiber feed-forward inhibition may be partially reduced with high frequency burst stimulation where feed-forward inhibition does not facilitate as strongly as excitation enabling excitation to dominate [59] (Fig 3) although this is not the case during development of the mossy fiber pathway [49]. Here, the remarkable finding is that cholinergic activation reduces mossy fiber feed-forward inhibition by >70% and noradrenergic activation by ~30% (Figs 1 and 2) reducing the attenuation of back-propagating action potentials and EPSPs (Fig 4), which is therefore predicted to facilitate LTP at CA3-CA3 recurrent synapses. The lesser effect of noradrenaline compared with acetylcholine may reflect less facilitation of LTP and ensemble formation although this is not explored in our spiking network models.

Further investigation revealed that not only does acetylcholine enable ensemble formation by reducing mossy fiber feed-forward inhibition, but it also alters the properties of the CA3 network to allow ensemble formation to occur rapidly and robustly with a high degree of overlap between ensembles. This supports findings in similar models of piriform cortex and CA3 attractor networks [28,51]. Enhancing cellular excitability within an attractor network such as the recurrent CA3 network increases the speed and robustness of synaptic plasticity due to increased spiking during ensemble activity (Fig 7). For stable ensemble formation and network configuration the increased excitability must be regulated by feedback inhibition (Fig 5). Theoretically, the reduction in CA3-CA3 synaptic efficacy caused by acetylcholine [28] might be predicted to reduce the efficiency of ensemble formation but our results show this is not the case (Fig 7) likely because of reduced interference between ensembles [51]. Furthermore, we found that this effect of acetylcholine enabled a greater overlap between ensembles whilst still maintaining their discrete identity (Fig 8). This is important for a couple of reasons: i) it allows an increased density of discrete ensembles to be encoded which in a finite network will increase its capacity to store information, and ii) an increase in overlap between ensembles has been suggested to facilitate memory consolidation and generalization during reactivation of ensembles that occurs during sleep {O'Donnell, 2014 #3203}. Interestingly, our model used a method to limit synaptic strengthening based on recently described STDP rules [41] coupled with a rate-based scaling factor, whereas previous models of similar autoassociative networks have used an LTP only rule with a saturation function [28,51]. Remarkably, both these methods produced very similar outcomes indicating that the effects of acetylcholine on the rate and degree of overlap for ensemble creation are independent of different plasticity rules. A more important factor may be non-linear dendritic conductances which increase the storage capacity for similar or overlapping memories within the CA3 network [106,107]. Future studies may determine how the mechanisms engaged by acetylcholine and non-linear conductances interact and combine within the hippocampal CA3 network.

A core symptom of Alzheimer's disease is deteriorating episodic memory, which may be ameliorated by treatment with cholinesterase inhibitors and is associated with degeneration of both noradrenergic fibers originating in locus coeruleus and cholinergic fibers originating in the basal forebrain. However, the mechanisms by which increasing the availability of acetylcholine in the brain provides this cognitive benefit remain obscure and it remains to be shown that

noradrenergic drugs could provide a similar benefit. At the behavioural level, our findings predict that cholinesterase treatment facilitates the formation of memory ensembles within the hippocampus and increases the storage capacity for separate memory representations whereas noradrenergic receptor activation will enhance memory ensemble formation without increasing storage capacity. It is widely reported that cholinesterase inhibitors provide cognitive enhancement [108] but the specific cognitive domains affected are less well characterized. At a network level, our findings predict that interventions to deprive the hippocampus of cholinergic or noradrenergic innervation will prevent the update of ensemble configurations in CA3 [109] and, furthermore, that stimulation of acetylcholine or noradrenaline release at specific locations will bias ensemble formation towards the incorporation of place cells representing those locations [18]. In addition, recent studies highlight the existence of 'event codes' in the hippocampus where the CA3 network forms generalised ensembles that may be read out in CA1 [91]. Based on our spiking neural network results, we predict that the formation of such event codes is dependent on cholinergic, but not noradrenergic signalling. As such, we hypothesize that mice trained on tasks that require the animal to generalise behaviour despite changing environments [91] will be sensitive to cholinergic antagonists administered during learning and should not form generalizable event codes, whereas mice treated with noradrenergic antagonists during learning should be unaffected. Manipulations of acetylcholine release in the hippocampus have largely focussed on the effects on oscillatory activity where acetylcholine has been found to promote theta activity and suppress sharp wave ripples [110]. However, in support of our predictions, cholinergic activation has also been found to increase the number of neurons incorporated into ensembles measured by their activity during sharp wave ripples [111].

Acetylcholine or noradrenaline release within the central nervous system has classically been portrayed as a signal for arousal and attention and is strongly associated with learning [13,112–115]. This model has been adapted to propose that acetylcholine and noradrenaline are released in response to arousal generated by different forms of uncertainty where acetylcholine is released in environments where the outcome is uncertain but known whereas noradrenaline is released in response to uncertainty that is unknown or unexpected [32]. In such a scenario new information needs to be incorporated into internal representations of CA3 ensembles to make the environment more familiar and the outcomes more predictable in situations of expected uncertainty signalled by acetylcholine [114]. In contrast, unexpected uncertainty signalled by noradrenaline requires a state change in representations (Sales et al., 2019) that requires formation of new ensembles in CA3 rather than adaptation of existing ones suggesting the importance of wider mechanisms external to the hippocampus to drive new input patterns from dentate gyrus. Our data reveal mechanisms whereby acetylcholine and noradrenaline can separately reconfigure the dentate gyrus and CA3 microcircuit to enable the formation of memory ensembles within the recurrent CA3 network according to varying behavioural imperatives.

## Materials and methods

### Ethical approval

All experiments were performed in accordance with the UK Animal Scientific Procedures Act (1986) and local guidance from the Home Office Licensing Team at the University of Bristol. The protocol was approved by the Animal Welfare and Ethics Review Board at the University of Bristol.

### Slice preparation

500 μm thick transverse slices of the dorsal hippocampus were prepared from 4–6 week old C57/BL6 mice. After cervical dislocation, brains were removed and submerged in ice-cold cutting solution saturated with oxygen (in mM: 85 NaCl, 75 Sucrose, 2.5 KCl, 25 Glucose, 1.25

NaH2PO4, 4 MgCl2, 0.5 CaCl2, 24 NaHCO3). Each hippocampus was dissected out and mounted onto a cube of agar then glued to the slicing plate such that hippocampi were positioned vertically and cut using a Leica VT1200 vibratome. Slices were then transferred to a holding chamber with oxygenated aCSF (in mM: 119 NaCl, 2.5 KCl, 11 Glucose, 1 NaH2PO4, 26.5 NaHCO3, 1.3 MgSO4, 2.5 CaCl2), incubated for 30 mins at 35˚C and left to rest for a further 30 minutes– 5 hours at room temperature.

## Electrophysiology

Slices recordings were made in a submerged recording chamber at 33–35˚C. CA3 pyramidal cells were visually identified using infra-red differential interference contrast on an Olympus BX-51W1 microscope. Recording pipettes with resistance 2–4 MΩ were pulled from borosilicate filamented glass capillaries and filled with a caesium-based intracellular solution (in mM: 130 CsMeSO3, 4 NaCl, 10 HEPES, 0.5 EGTA, 10 TEA, 1 QX-314 chloride, 2 MgATP, 0.5 NaGTP). For recordings of CA3 pyramidal membrane potential and input resistance a potassium-based recording solution was used (in mM: 120 KMeSO3, 8 NaCl, 10 KCl, 10 HEPES, 4 MgATP, 0.3 NaGTP, 0.2 EGTA). Series resistances were continuously monitored and recordings discarded if series resistance > 25 MΩ or changed by >50%. Recordings were collected using a Multiclamp 700A amplifier (Molecular Devices) filtered at 4 kHz and sampled at 10 or 25 kHz using Signal or Spike2 acquisition software, and a CED Power 1401 data acquisition board.

Postsynaptic currents in the mossy fiber pathway were evoked by placing monopolar stimulation electrodes in the granular layer of the dentate gyrus and applying 200 µs pulses driven by a Digitimer DS2A Isolated stimulator. Stimulating the granular layer avoids the risk of contamination from perforant path inputs, associational/commissurals, or monosynaptic inhibitory synapses. Excitatory currents were obtained by holding the cell in voltage-clamp mode at -70 mV, while inhibitory currents were obtained by holding at +10 mV. Candidate mossy fiber driven responses were chosen based on their latency (2–3 ms for excitatory, 5–10 ms for inhibitory). Excitatory currents were also selected based on kinetics (< 1 ms 20–80% rise time) and short-term facilitation in response to 4 pulses at 20 Hz. Stimulation strength was calibrated to the minimum strength that evoked a response (minimal stimulation). Excitatory and inhibitory responses were blocked > 80% by 1 µM DCG-IV in 100% cases (Fig 1; n = 9 MF-EPSCs, n = 6 MF-IPSCs), therefore it can be inferred that this approach reliably stimulated mossy fibers [54]. Associational/commissural synaptic responses were evoked by placing a bipolar stimulating electrode in stratum radiatum and stimulation intensities set to evoke synaptic currents ~100-200pA amplitude and therefore multiple axons. Carbachol (CCh) or noradrenaline (NA) were bath applied for 5–10 minutes before measuring effect on synaptic transmission. Effects of CCH or NA were normalised to control values taken in first 3 minutes of the experiment.

Experimental unit was defined as cell with only one cell recorded per slice. Up to 2 cells were recorded from each animal. Measurements were taken as an average of 3 responses to obtain a data point per min, averages represent mean ± S.E.M. Time series data were normalised to the last 5 min of baseline and drug effects were assessed by comparing the average PSC amplitudes during the last 5 min of application. Data were analysed using a paired two-tailed Student's t-test. The results of these t-tests are also represented as asterisk on summary histograms of the average drug effect. Significance assigned * if $p < 0.05$, ** if $p < 0.01$ and *** if $p < 0.001$. Power analysis indicated that minimum sample size n = 6 for mossy fiber PSCs and n = 4 for associational/commissural PSCs was required to distinguish drug effects at 95% confidence intervals with 80% power using effect sizes and variability calculated from our previous data. Data were processed, analysed and presented using Signal (CED) v5.12, Matlab (R2019a) and Graphpad Prism v8.

## Short-term plasticity model

The Tsodyks-Markram model [116,117] was adopted due to its widespread use, simplicity, and relation to biophysics. This model captures pre-synaptic release dynamics with two variables, a facilitating process $f$ and a depressing process $d$, that represent 'resources' available to drive synaptic transmission. These evolve as follows:

$$\frac{df}{dt} = \frac{f_0 - f}{\tau_f} + a(1-f)\sum_s \delta(t-t_s) \tag{11}$$

$$\frac{dd}{dt} = \frac{1-d}{\tau_d} - fd\sum_s \delta(t-t_s) \tag{12}$$

The dynamics of $f$ are governed by three parameters: $f_0$ (baseline value of $f$), $\tau_f$ (decay time constant of $f$), and $a$ (increment scaling factor with incoming spike at time $t_s$). This variable loosely represents the build-up of free calcium ions in the presynaptic terminal that triggers exocytosis of neurotransmitter-containing vesicles into the synaptic cleft. Dynamics of $d$ is governed by a single parameter $\tau_d$, and loosely represents the availability of docked vesicles at release sites. Since these variables are bounded between 0 and 1, to convert into a conductance amplitude these are multiplied by a conductance scaling factor $g$, i.e.,

$$\text{PSC Amplitude} = gfd \tag{13}$$

This basic model can be extended to more complex facilitation models for example by making parameters $f_0$ and $a$ time dependent (Hennig, 2013), i.e.,

$$\frac{df_0}{dt} = \frac{\tilde{f}_0 - f_0}{\tau_{f_0}} + b_{f_0}(1-f_0)\sum_s \delta(t-t_s) \tag{14}$$

$$\frac{da}{dt} = \frac{a_0 - a}{\tau_a} + b_a(1-a)\sum_s \delta(t-t_s) \tag{15}$$

Where $\tilde{f}_0$ is a baseline for $f_0$, $a_0$ is the baseline for $a$, $\tau_{f_0}$ is the time constant for $f_0$, $\tau_a$ is the time constant for $a$, $b_{f_0}$ is the increment scaling factor for $f_0$, and $b_a$ is the increment scaling factor for $a$. It can also be reduced by making $f$ or $d$ constant. Since the mossy fiber synapse is well known for its large pool of readily releasable vesicles that can be quickly replenished, a simple reduction is to keep $d$ constant at 1, which is true when $\tau_d << \min(ISI)$. Further complexity can be incorporated by allowing multiple independent depressing variables, or by having a time dependent scaling factor

$$\frac{dg}{dt} = -g/\tau_g + k\sum_s \delta(t-t_s) \tag{16}$$

where $\tau_g$ is the time constant for short-term changes in conductance, and $k$ is an increment scaling factor that can take positive values for facilitation, or negative values for depression.

## Naturalistic stimulation patterns

Previous research has shown that short-term facilitation models are difficult to constrain with responses evoked by regular stimulation protocols [60], and that irregular or naturalistic stimulus trains allowed much better fits to data due to sampling across a broader range of inter-

stimulus intervals (ISI) [60,118]. Dentate gyrus granule cells in vivo have been shown to have bimodal ISI distributions, with long periods of quiescence punctuated by short bursts of action potential firing [52,119]. This bimodal ISI distribution was modelled as a doubly stochastic Cox process to allow generation of stimuli resembling natural spike patterns [120]. Each Cox process $i$ is defined by a rate parameter $\lambda_i$, and a refractoriness parameter $\sigma_i$. These two processes are then mixed with responsibility $\pi$ i.e.,

$$P(ISI) = \pi q_1 + (1 - \pi)q_2 \tag{17}$$

$$q_i = r_i \text{ if } r_i > x_i \text{ else } = 0 \tag{18}$$

$$x_i \sim \mathcal{N}(\mu_i, \sigma_i^2) \tag{19}$$

$$r_i \sim \text{Exp}(\lambda_i) \tag{20}$$

$$\pi \sim \text{Ber}(p) \tag{21}$$

In brief, ISIs were generated by sampling from two exponential distributions with parameters $\lambda_i$, which were rejected if they were less than a sample from a normal distribution with standard deviation $\sigma_i$. These candidate ISIs were then accepted according to a Bernoulli distribution with probability $\pi$. These parameters were set as $\lambda_1 = 3.0$ Hz, $\lambda_2 = 0.25$ Hz, $\mu_1 = 0.006$ s, $\mu_2 = 0.0025$ s, $\sigma_1 = 0.12$ s, $\sigma_2 = 0.01$ s, $\pi = 0.55$. Ninety-nine ISIs were sampled to provide 100 spike times for a stimulation protocol lasting 525 seconds.

## Model fitting

The Bayesian parameter inference procedure used by [60] was used to fit parameters to the model. Each model was considered in an iterative form, integrating $f$ and $d$ over each ISI $\Delta t_s$ between the $n^{th}$ and $n+1^{th}$ spike to output a normalised post-synaptic conductance amplitude for the $n^{th}$ spike in the sequence, i.e.,

$$f_{n+1} = f_0 - (f_0 - f_+)\exp(-\Delta t_s/\tau_f), \ f_+ = f_n + a(1 - f_n) \tag{22}$$

$$d_{n+1} = 1 - (1 - d_+)\exp(-\Delta t_s/\tau_d), \ d_+ = d_n(1 - f_n) \tag{23}$$

from the original Tsodyks-Markram formalism, however for more complex models in which $a$ and $f_0$ are time dependent

$$f_{0,n+1} = \tilde{f}_0 - (\tilde{f}_0 - f_{0,+})\exp(-\Delta t_s/\tau_{f_0}), f_{0,+} = f_{0,n} + b_{f_0}(1 - f_{0,n}) \tag{24}$$

$$a_{n+1} = a_0 - (a_0 - a_+)\exp(-\Delta t_s/\tau_a), a_+ = a_n + b_a(1 - a_n) \tag{25}$$

Additionally, when $f_0$ is time-dependent, Eq 21 becomes

$$
\begin{aligned}
f_{n+1} = \frac{1}{\tau_{f_0} - \tau_f}&(-\tau_{f_0}\tilde{f}_0(\exp(-\Delta t/\tau_{f_0}) - 1) + \tau_f\tilde{f}_0(\exp(-\Delta t/\tau_f) - 1) \\
&+ \exp(-\Delta t/\tau_f)(\tau_{f_0}f_+ - \tau_{f_0}f_{0,+} - \tau_f f_+) + \tau_{f_0}f_{0,+}\exp(-\Delta t/\tau_{f_0})
\end{aligned} \tag{26}
$$

These were then compared to PSC amplitude estimated by the difference between response peak and a baseline taken just before stimulus onset.

Since these equations are deterministic, a likelihood model was constructed where the amplitude was used as parameters for a normal distribution, i.e.,

$$P(D|\theta) = \mathcal{N}(A, A/2) \tag{27}$$

where $A$ is PSC amplitude (Eq 13).

Exponential priors were used for conductance scaling parameters, beta priors for baseline and increment parameters, and uniform priors for time constants. This choice was made in order to bias conductances towards smaller values and keep baselines low as would be expected from facilitating synapses. Posterior distributions were estimated using Markov Chain Monte Carlo sampling via the Metropolis-Hastings algorithm using the pymc python module. We subsequently examined the covariance structure of samples in the resulting Markov chain to investigate dependencies amongst parameters [65].

Model selection was conducted using AIC and BIC weights [63], a transformation of AIC and BIC values into a probability space, with best model having the highest weight. This made it possible to compare the best fitting model over the population when fitted for each individual sample. For the information criterion of a model for a given sample $C_n^m$ this is defined as

$$w(C) = \frac{\exp(-0.5\Delta_i(C))}{\sum_{k=1}^{K} \exp(-0.5\Delta_k(C))} \tag{28}$$

where

$$\Delta(C) = \frac{1}{N} \sum_n |C_n - \arg\min_m(C^m)| \tag{29}$$

Goodness-of-fit for the best fitting model was assessed by estimating Bayesian poster-predictive $p$-values, where samples are drawn from the posterior-predictive distribution and discrepancies $D(x|\theta)$ to expected values $e$ from the model $x_{sim}$ and to the data $x_{obs}$ are compared [121], i.e.,

$$p = \Pr[D(x_{sim}|\theta) > D(x_{obs}|\theta)] \tag{30}$$

where

$$D(x|\theta) = \sum_j \left(\sqrt{x_j} - \sqrt{e_j}\right)^2 \tag{31}$$

If discrepancies were similar, i.e., $0.025 < p < 0.975$, then the model is assessed to fit well. This quantifies how easy it is to discriminate between posterior samples and actual data.

Best fit Tsodyks-Markram model parameters for the $f_2$ model of EPSC short-term plasticity and the afd model of IPSC short-term plasticity are given in Table 1 together with the parameter changes incurred by noradrenaline or acetylcholine. These parameter sets were used for the simulations in Fig 3 to determine Excitation–Inhibition ratios across a range of activity patterns.

## Compartmental modelling

15 multi-compartment reconstructions (number of compartments ranged from 943 to 2110) of CA3 pyramidal cells with active dendrites [70,71] were used to study whether mossy fiber feed-forward inhibition could regulate action potential back-propagation, which would be necessary for feed-forward inhibition to regulate plasticity between recurrent synapses. Simulations were carried out using NEURON.

Active conductances included voltage-gated sodium ($Na_V$), voltage-activated potassium conductance including delayed rectifier ($K_{DR}$), M-current ($K_M$), fast-inactivating A-type ($K_A$), calcium conductances including N-type ($Ca_N$), T-type ($Ca_T$), and L-type ($Ca_L$), calcium-activated potassium conductances ($K_C$ and $K_{AHP}$). Calcium extrusion was modelled as a 100ms decay to a resting $Ca^{2+}$ of 50 nM. Channel kinetics were similar to those used in other hippocampal pyramidal neuron models [122].

Somatic compartments contained all conductances, dendritic compartments contained all except $K_M$, and the axonal compartment contained only $Na_V$, $K_{DR}$, and $K_A$. For action potential generation, sodium conductance was five times higher in the axon than the rest of the neuron. Conductances (in $\mu S/cm^2$) followed those produced by Hemond et al (2008) to set the cell to respond to current injection with burst firing that is canonical to CA3 pyramidal neurons [123,124]: $gNa_V$ = 0.022, $gK_{DR}$ = 0.005, $gK_M$ = 0.017, $gK_A$ = 0.02, $gCa_N$ = 0.00001, $gCa_T$ = 0.00001, $gCa_L$ = 0.00001, $gK_C$ = 0.00005, $gK_{AHP}$ = 0.0001.

Dendritic compartments along the apical dendrite were subdivided according to distance (in microns from soma) into those within stratum lucidum ($\leq 150$), stratum radiatum ($>150$ or $\leq 400$), and stratum lacunosum moleculare ($>400$). Mossy fiber synapses were targeted towards compartments in stratum lucidum. Feed-forward inhibition was targeted towards somatic compartments (50%) and dendritic compartments in stratum lucidum and stratum radiatum (50%) reflecting the diversity of interneuron subtypes and their targets [45].

Synaptic input was modelled using bi-exponential kinetics and short-term plasticity dynamics fit to experimental data. The effect of carbachol was modelled as a three-fold reduction in feed-forward inhibitory conductance.

## CA3 network modelling

The hippocampal CA3 region was modelled as a small all-to-all recurrent network comprised of excitatory and inhibitory point neurons with adaptive quadratic-integrate-and-fire dynamics with parameters to reflect the firing patterns in response to current injection of CA3 pyramidal cells and fast-spiking basket cells respectively [27,76]. Continuous membrane dynamics for neuron $i$ is described by two equations:

$$C_m \frac{dv_i}{dt} = k(v_i - v_r)(v_i - v_t) - u_i - g_E(v_i - v_E) - g_I(v_i - v_I) \tag{32}$$

$$\frac{du_i}{dt} = a[b(v_i - v_r) - u_i] \tag{33}$$

These governed dynamics until $v_i \geq v_{peak}$, at which time $v_i$ and $u_i$ were reset to

$$v_i \leftarrow c \tag{34}$$

$$u_i \leftarrow u_i + d \tag{35}$$

where $v_i$ describes membrane potential, and $u_i$ is a slow adaptation variable. As in [27], parameters for excitatory cells were: $C_m$ = 24 pF, $k$ = 1.5 pA/mV$^2$, $a$ = 10 Hz, $b$ = 2 nS, $c$ = -63 mV, $d$ = 60 pA, $v_{rest}$ = -75 mV, $v_t$ = -58 mV, $v_{peak}$ = 29 mV. We chose parameters for inhibitory cells to produce fast spiking behaviour described by [27]: $C_m$ = 16 pF, $k$ = 1.5 nS/mV, $a$ = 900 Hz, $b$ = 2 nS, $c$ = -80 mV, $d$ = 400 pA, $v_{rest}$ = -65 mV, $v_t$ = -50 mV, $v_{peak}$ = 28 mV.

Synaptic reversal potentials were set as $v_E$ = 10 mV, and $v_I$ = -80 mV.

Excitatory and inhibitory cells were connected through four types of synapses: excitatory to excitatory cell (EE) synapses, excitatory to inhibitory cell synapses (EI), inhibitory to excitatory

cell synapses (IE), and inhibitory to inhibitory cell (II) synapses. Kinetics were modelled as exponential synapses such that the synaptic conductance $g_{syn}$ evolved according to:

$$\frac{dg_{syn}}{dt} = -g_{syn}/\tau_g + \tilde{g_{syn}} w \sum_s \delta(t - t_s) \tag{36}$$

where $\tilde{g_{syn}}$ is the maximum synaptic conductance (0.5 nS for excitatory synapses and 1.0 nS for inhibitory synapses), $\tau_g$ = 10 ms for EE and EI synapses and 20 ms for IE and II synapses [125], and $w$ is the synaptic weight determined by STDP rules for EE and IE synapses, and fixed at 0.3 nS otherwise. Cholinergic modulation was implemented by changing a subset of network parameters as described by [27] whereas noradrenaline did not cause any parameter changes (Table 2).

EE and IE synapses were subject to spike timing-dependent plasticity of the form

$$\Delta w = \eta[\exp(-|t_{pre} - t_{post}|) - z] \tag{37}$$

For all pre-post pairs where $\eta$ is a learning rate, and $z$ is a scaling factor. This provides a symmetric STDP rule used previously as a homeostatic means to balance excitation and inhibition through inhibitory plasticity [125] and has also recently been shown to operate at CA3 associated commissural synapses [41]. In the case of IE synapses, $\eta$ and $z$ are fixed, however in the case of EE synapses these evolve according to

$$\frac{d\eta}{dt} = -\eta/\tau_\eta + \xi \sum_{t_{post}} \delta(t - t_{post}) \tag{38}$$

$$\frac{dz}{dt} = -z/\tau_z + \rho_{max}^{-1} \sum_{t_{post}} \delta(t - t_{post}) \tag{39}$$

This makes $\eta$ and $z$ track the postsynaptic firing rate for two different purposes, and on different time scales since $\tau_\eta$ = 100 ms, and $\tau_z$ = 1 second; $\eta$ becomes a burst detector that increases the learning rate by a factor $\xi$ (equal to 0.02) meaning STDP requires multiple postsynaptic spikes to be activated, and $z$ scales STDP such that the postsynaptic firing rate reaches a maximum $\rho_{max}$ (10 Hz), more pre- and post-synaptic spike pairs cause depression, preventing STDP from inducing unrealistically high firing rates in an excitatory recurrent network. Additionally, STDP was bounded between 0 and the maximum conductance of the synapse.

Synaptic input to cells was comprised of recurrent and feed-forward inputs, i.e.,

$$g_E^i = \sum_{ij} g_{EY}^{ij} + \sum_{ik} g_{FF}^{ik} \tag{40}$$

$$g_I^i = \sum_{ij} g_{IY}^{ij} \tag{41}$$

where $g_{FF}$ is the conductance of feed-forward input, $g_{EY}$ is excitatory recurrent input, and $g_{IY}$ is recurrent inhibition. Feed-forward input was given only to excitatory cells. CA3 Network architecture and feed-forward dynamics varied according to each simulation.

Network retrieval performance is measured by a discrimination index that assumes ensemble population rates are read by a downstream neuron. The greater the difference in ensemble population rates, the easier it is to differentiate between ensembles and retrieval is more

precise. The discrimination index $D$ is defined as

$$D = \frac{\int v\_\gamma(t)dt}{\sum_{i=\{-1,0,+1\}} \int v_{\gamma+i}(t)dt} \tag{42}$$

$$v(t) = \frac{\int K(\tau)\sum_i S_i(t-\tau)d\tau}{\int K(\tau)d\tau} \tag{43}$$

$$S(t) = \sum_s \delta(t-t_s) \tag{44}$$

where $v_\gamma(t)$ is the population rate $v$ of ensemble $\gamma$, $S_i(t)$ is the spike train $S(t)$ of neuron $i$, $K(t)$ is a kernel averaging the spike train over a defined window, and $\delta(t-t_s)$ is the delta function modelling a spike at time $t_s$. This discrimination index essentially calculates the ratio *signal/ (signal + noise)* where the signal is the population rate of the ensemble representing the memory being retrieved, and the noise is the population rate of ensembles representing memories that should not be retrieved and are interfering with the retrieval process. As such, when $D$ is smaller the interference from neighbouring neurons is higher.

## Supporting information

**S1 Fig.** A-B) Latency, rise times, and jitter of mossy fiber driven EPSCs (n = 11) (A) and IPSCs (n = 12) (B). C-D) Reversal potential estimation of glutamatergic (n = 5) (C) and GABAergic (n = 6) (D) transmission at CA3 pyramidal cells. E-F) Spontaneous EPSC (n = 6) (E) and IPSC (n = 5) (F) frequency recorded before and after carbachol application. G-H) CA3 pyramidal cell input resistance (n = 5) (G) and holding current at -70 mV (n = 5) (H) before and after carbachol application. I) Dose-response of carbachol effect on IPSC amplitudes (n = 3).
(PDF)

**S2 Fig.** A-B) Goodness-of-fit for EPSC (A) and IPSC (B) short-term plasticity models assessed by Bayesian posterior predictive *p*-values. Plots show the observed data for all experiments (EPSCs and IPSCs respectively) together with the expected values from the model before and after the application of 5 μM carbachol. *p*-values close to 0.5 indicate best fit. C-D) Posterior distributions for parameters of best fitting models given data for EPSCs (C) and IPSCs (D).
(PDF)

**S3 Fig.** A) Covariance in excitatory short-term plasticity model parameters (mean +/- sem). B) Covariance in inhibitory short-term plasticity model parameters (mean +/- sem). C) Example pairwise distributions of excitatory short-term plasticity model parameters D) Example pairwise distributions of inhibitory short-term plasticity parameters.
(PDF)

**S4 Fig. CA3 pyramidal cell morphologies used for biophysical modelling.** A) All 15 cell morphologies plotted from NEURON spatial information. B) For each cell morphology, back-propagating action potential amplitude before (left) and after (middle) cholinergic modulation, and the difference in amplitude (right) are shown distributed across each CA3 pyramidal cell.
(PDF)

**S5 Fig.** A) Slices of data shown in Fig 7C along frequency (top) and time (bottom) axes. B) Slices of data shown in Fig 8C along overlap (top) and time (bottom) axes. Colour coding for

plots is indicated in the legend representing inclusion of different effects of acetylcholine in CA3.
(PDF)

**S6 Fig.** Model CA3 spiking activity (left) and resulting pyramidal cell weight matrix (right) with mossy fiber bursting at 30 Hz (A), 20 Hz (B), and 20 Hz with cholinergic modulation (C). (PDF)

**S7 Fig.** Model CA3 spiking activity and resulting weight matrix without cholinergic modulation (A) and with cholinergic modulation (B) with 0 (top), 2 (middle), and 4 (top) cells overlapping between ensembles.
(PDF)

## Acknowledgments

We thank members of the Mellor lab for helpful discussion and C. O'Donnell and M. Ashby for comments on previous versions of the manuscript.

## Author Contributions

**Conceptualization:** Luke Y. Prince, Jack R. Mellor.

**Formal analysis:** Luke Y. Prince.

**Funding acquisition:** Jack R. Mellor.

**Investigation:** Luke Y. Prince, Travis Bacon, Rachel Humphries.

**Methodology:** Luke Y. Prince, Krasimira Tsaneva-Atanasova, Claudia Clopath.

**Project administration:** Jack R. Mellor.

**Supervision:** Krasimira Tsaneva-Atanasova, Claudia Clopath, Jack R. Mellor.

**Writing – original draft:** Luke Y. Prince, Krasimira Tsaneva-Atanasova, Claudia Clopath, Jack R. Mellor.

**Writing – review & editing:** Luke Y. Prince, Travis Bacon, Rachel Humphries, Krasimira Tsaneva-Atanasova, Claudia Clopath, Jack R. Mellor.

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
