## [Decision Letter · Decision Letter 0]

4 Jan 2021

Dear Dr. Mellor,

Thank you very much for submitting your manuscript "Separable actions of acetylcholine and noradrenaline on neuronal ensemble formation in hippocampal CA3 circuits" for consideration at PLOS Computational Biology.

As with all papers reviewed by the journal, your manuscript was reviewed by members of the editorial board and by several independent reviewers. In light of the reviews (below this email), we would like to invite the resubmission of a significantly-revised version that takes into account the reviewers' comments.

In particular, the authors should make more of an effort to justify the specific choice of parameter values used in their simulations, explore how their qualitative findings depend on these specific parameter values, and describe the specific predictions of this work for future experiments and our functional understanding of the hippocampal memory system.

We cannot make any decision about publication until we have seen the revised manuscript and your response to the reviewers' comments. Your revised manuscript is also likely to be sent to reviewers for further evaluation.

Sincerely,

Daniel Bush

Associate Editor

PLOS Computational Biology

Kim Blackwell

Deputy Editor

PLOS Computational Biology

In particular, the authors should make more of an effort to justify the specific choice of parameter values used in their simulations, explore how their qualitative findings depend on these specific parameter values, and describe the specific predictions of this work for future experiments and our functional understanding of the hippocampal memory system

Reviewer's Responses to Questions

**Comments to the Authors:**

Reviewer #1: It has long been believed that neuromodulation acts as a gatekeeper of the hippocampal memory system, deciding which aspects of experience are novel/important enough to warrant memory storage. The mechanisms of this process are however not completely well understood: for instance both acetylcholine and noradrenaline can affect the input into the system (via the DG-CA3 mossy fiber connection), but their relative contribution to memory gating is unclear. This manuscript uses slice electrophysiology to jointly characterize the role of acetylcholine (Ach) and noradrenaline (NA) on the mossy fiber excitatory input and di-synaptic feedforward inhibition. These findings are incorporated into computational models of ensemble formation to gauge their functional implications.

ACH was shown to induce a lasting reduction in the amplitude of mossy fiber EPSPs. It also caused a reduction of IPSPs, while also reducing facilitation ratios. Thirdly, it increased the excitability of CA3 cells. For NA, the applied dose had no effect on EPSCs and a weaker effect of IPSCs. Careful model comparison also revealed differences w.r.t. short term plasticity. Ach lead to much bigger shifts in EI balance than NA, with potentially important implications to CA3 recurrent synapses plasticity, as evidenced by compartmental models.

These processes are then incorporated into a spiking recurrent network to investigate ensemble formation, where ensembles here are defined somewhat extremely (c.f. lines 469-70) as fully connected components completely decoupled from the rest of the network. This model is uses to iteratively investigate the contribution of different components (mostly numerically)

Major:

While the numerical results are clear, there seem to be a lot of knobs in the modeling approach (and to some degree the experiment), which make it hard to infer the generality of the conclusions; I would have appreciated a bit more more explanation on how closely does the modeling setup reflect known physiology, and how robust the results are to changing different parameters.

The order in which the information is presented is a bit odd in places, with results being anticipated and then repeated in multiple sections and going back and forth between experiments and models, making the individual points not as sharp as they could be, see also detailed comments below. A more effective narrative thread would be to 1) identify all the different immediate effects of different neuromodulators, then 2) put them together in a circuit model to predict differences in ensemble formation, and lastly 3) take the different effects apart to reveal which consequence of neuromodulation has most effect functionally.

I would have liked some concrete predictions coming from the model; for instance w.r.t. effects of systemic neuromodulator manipulation on memory ability of animals — I would think that the generalization ability for different memories may also be affected (going with the allowable degree of overlap across ensembles).

Minor:

Somehow confounding how the change in E(I)PSC amplitude and depression/facilitation are talked about w.r.t to fig 1 and 2. The title suggests that short term plasticity should be referred to in fig 2 and associated text but is already mentioned w.r.t. fig 1. Overall, some of the text felt repetitive. There may be a better way to streamline the presentation of the two...

don’t get the logic of the statement in line 219: the EI ratio changes seem quite dramatically different between Ach and NA manipulations even at the raw data level, why say they could be similar? In fact the results contradict it.

Line 330-331: The effects in inhibition seem contradictory: high firing rates and strong synaptic depression seems like a particularly energy inefficient way of neurally implementing a release from inhibition. Rather it would indicate a shift towards different patterns of spike timing…

Line 495: “a colored noise signal” feels odd; maybe just ‘colored noise’ (?)

L536: the role of Ach on excitability has been mentioned in the results before, rather apologetically, leaving the reader to wonder whether it was due to simply a release in inhibition or also involved changes in intrinsic neural properties. Now we go back to experiments and measure that in the middle of setting up possible coding properties of the Ach manipulation; I feel the narrative would be better served if the experimental results were more coherently explained and closer together in the storyline.

L590: “applying the CA3 network effects” phrasing odd

The link to neuromodulators signaling different forms of uncertainty is out of place with the rest of the ensemble formation discussion, I’d try to integrate it better.

Reviewer #2: uploaded as an attachment

**Have all data underlying the figures and results presented in the manuscript been provided?**

Reviewer #1: Yes

Reviewer #2: **No: **I could not find Table 1

PLOS authors have the option to publish the peer review history of their article (what does this mean?). If published, this will include your full peer review and any attached files.

Reviewer #1: No

Reviewer #2: No
---

## [Decision Letter · Decision Letter 1]

28 May 2021

Dear Dr. Mellor,

Thank you very much for submitting your manuscript "Separable actions of acetylcholine and noradrenaline on neuronal ensemble formation in hippocampal CA3 circuits" for consideration at PLOS Computational Biology. As with all papers reviewed by the journal, your manuscript was reviewed by members of the editorial board and by several independent reviewers. The reviewers appreciated the attention to an important topic. Based on the reviews, we are likely to accept this manuscript for publication, providing that you modify the manuscript according to the review recommendations.

In particular, this paper deals with a complex combination of experiments and detailed modelling, and the exact combination of protocols and parameters that give rise to each finding are not always immediately obvious to the reader. It is crucial that sufficient detail is provided to allow these simulations (and their results) to be replicated. Hence, I encourage the authors to make a concerted effort to simplify, clarify and improve the comprehensibility of the methods for each set of simulations. I would also strongly recommend that their code is made available online, to further this aim.  Also, the policy of PLoS Computational Biology is that code must be made publicly available.

Sincerely,

Daniel Bush

Associate Editor

PLOS Computational Biology

Kim Blackwell

Deputy Editor

PLOS Computational Biology

[LINK]

Reviewer's Responses to Questions

**Comments to the Authors:**

Reviewer #1: I find the revision addresses all of the concerns I had previously raised. I still think the overall structure of the narrative could be improved by some reordering, but I get why the authors chose to keep things as they were.

Reviewer #2: text uploaded

**Have the authors made all data and (if applicable) computational code underlying the findings in their manuscript fully available?**

Reviewer #1: **No: **I may have missed it, but I did not see any statement promising release of code or data.

Reviewer #2: **No: **Not only data but also the code should become avalibale in a public repository.

PLOS authors have the option to publish the peer review history of their article (what does this mean?). If published, this will include your full peer review and any attached files.

Reviewer #1: No

Reviewer #2: No

Figure Files:

Data Requirements:

Reproducibility:

References:

---

## [Decision Letter · Decision Letter 2]

12 Aug 2021

Dear Dr. Mellor,

Thank you very much for submitting your manuscript "Separable actions of acetylcholine and noradrenaline on neuronal ensemble formation in hippocampal CA3 circuits" for consideration at PLOS Computational Biology. As with all papers reviewed by the journal, your manuscript was reviewed by members of the editorial board and by several independent reviewers. The reviewers appreciated the attention to an important topic. Based on the reviews, we are likely to accept this manuscript for publication, providing that you modify the manuscript according to the review recommendations.

Specifically, if you could clarify the derivation of Equations 7-9 - as discussed by Reviewer 2 - and make the other minor corrections they suggested, that would be greatly appreciated. 

Sincerely,

Daniel Bush

Associate Editor

PLOS Computational Biology

Kim Blackwell

Deputy Editor

PLOS Computational Biology

[LINK]

Reviewer's Responses to Questions

**Comments to the Authors:**

Reviewer #2: I the second revision, all of my comments have been addressed. The authors should nevertheless check this:

The derivation of equations 7-9 was discussed in the revision. I now understand that delta_t_within could be neglected if it is small compared to delta_t_between. However, I still do not see why the number of spikes within a burst can be neglected. I could derive, for example, eq. 9 by assuming that there is one spike per burst. The explanation in the point-by-point response that "by the time of the burst, the effect of the previous burst is negligible" does not make sense to me because the anticipated steady-state values should depend on the structure of the bursts, which are characterized by several spikes.

Equation number "26" does not exist, instead it has number "116".

Please check again the vertical scales in Figs 4F and G. In the point-by-point response it was said that a factor 5 was missing, but the figures were not updated.

The caption of Fig. 5 still shows "Ppost" instead of the symbol \\rho_post

In Table 2, the bottom left entry shows just "~"

**Have the authors made all data and (if applicable) computational code underlying the findings in their manuscript fully available?**

Reviewer #2: Yes

PLOS authors have the option to publish the peer review history of their article (what does this mean?). If published, this will include your full peer review and any attached files.

Reviewer #2: No

Figure Files:

Data Requirements:

Reproducibility:

References:

---

## [Editor Report · Decision Letter 3]

8 Sep 2021

Dear Dr. Mellor,

We are pleased to inform you that your manuscript 'Separable actions of acetylcholine and noradrenaline on neuronal ensemble formation in hippocampal CA3 circuits' has been provisionally accepted for publication in PLOS Computational Biology.

Best regards,

Daniel Bush

Associate Editor

PLOS Computational Biology

Kim Blackwell

Deputy Editor

PLOS Computational Biology

---

## [Editor Report · Acceptance letter]

21 Sep 2021

PCOMPBIOL-D-20-02070R3 

Separable actions of acetylcholine and noradrenaline on neuronal ensemble formation in hippocampal CA3 circuits

Dear Dr Mellor,

I am pleased to inform you that your manuscript has been formally accepted for publication in PLOS Computational Biology. Your manuscript is now with our production department and you will be notified of the publication date in due course.

With kind regards,

Andrea Szabo
